# A new global ice sheet reconstruction for the past 80 000 years

Evan J. Gowan [1,2 ✉], Xu Zhang [3,4], Sara Khosravi[5], Alessio Rovere [2], Paolo Stocchi[6], Anna L. C. Hughes [7,8], Richard Gyllencreutz [9], Jan Mangerud [8], John-Inge Svendsen[8] & Gerrit Lohmann [1,2]

The evolution of past global ice sheets is highly uncertain. One example is the missing ice problem during the Last Glacial Maximum (LGM, 26 000-19 000 years before present) – an apparent 8-28 m discrepancy between far-field sea level indicators and modelled sea level from ice sheet reconstructions. In the absence of ice sheet reconstructions, researchers often use marine $\delta^{18}O$ proxy records to infer ice volume prior to the LGM. We present a global ice sheet reconstruction for the past 80 000 years, called PaleoMIST 1.0, constructed independently of far-field sea level and $\delta^{18}O$ proxy records. Our reconstruction is compatible with LGM far-field sea-level records without requiring extra ice volume, thus solving the missing ice problem. However, for Marine Isotope Stage 3 (57 000-29 000 years before present) - a pre-LGM period - our reconstruction does not match proxy-based sea level reconstructions, indicating the relationship between marine $\delta^{18}O$ and sea level may be more complex than assumed.

[1] Alfred Wegener Institute, Helmholtz Center for Polar and Marine Research, Bremerhaven, Germany. [2] MARUM, University of Bremen, Bremen, Germany. [3] Center for Pan Third Pole Environment (Pan-TPE), Key Laboratory of Western China's Environmental Systems, (Ministry of Education), College of Earth and Environmental Science, Lanzhou University, Lanzhou, China. [4] CAS Center for Excellence in Tibetan Plateau Earth Sciences, Chinese Academy of Sciences (CAS), Beijing, China. [5] Alfred Wegener Institute, Helmholtz Center for Polar and Marine Research, Potsdam, Germany. [6] NIOZ, Texel, The Netherlands. [7] Department of Geography, University of Manchester, Manchester, UK. [8] Department of Earth Science, University of Bergen and Bjerknes Centre for Climate Research, Bergen, Norway. [9] Department of Geological Sciences, Stockholm University, Stockholm, Sweden. ✉email: evangowan@gmail.com

Glacial isostatic adjustment (GIA)-based ice sheet reconstructions are commonly used for paleoclimate modeling and assessing present-day Earth deformation and sea-level change. As a result, it is essential that the modeled history of the ice sheets adheres to local geological and geophysical constraints as much as possible. Prior reconstructions of ice sheets have been created by using methods that try to fit GIA constraints without regards to ice physics[1], using a regional thickness scaling parameter on an originally glaciologically constrained model[2], or by averaging together the results of thousands of low-resolution ice sheet model simulations[3]. All of these reconstructions use assumptions on far-field sea level as part of their tuning strategy. In the period prior to the LGM, proxy records based on oceanic $\delta^{18}O$ fluctuations are often used to tune reconstructions when there is an absence of paleo sea-level data[1,4].

Recent work in the Hudson Bay region in North America[5–7] shows that during part of MIS 3, ice-free conditions may have existed. Under this interpretation, the climatic conditions in this area were favorable to allow the growth of forests, with a climate that was potentially analogous to present[8]. This would indicate that not only was the Laurentide Ice Sheet reduced in size but it also had to be far enough removed from southern Hudson Bay to not strongly affect the climate there. Pico et al.[9], through GIA modeling, provided additional support for a reduced extent Laurentide Ice Sheet to explain high relative MIS 3 sea-level indicators along the eastern coast of the United States. The dating methods used for inferring reduced ice sheet extent are near the limit of their reliability during mid-MIS 3[10]. If regarded as minimum ages, then these deposits could be from an earlier ice-free period, such as the last interglacial.

Our reconstruction, called PaleoMIST 1.0 (Paleo Margins, Ice Sheets, and Topography), is created independently of indirect proxy records and far-field sea-level records. This allows us to investigate two of the most contentious problems when assessing past ice sheet configuration and sea level. First, our reconstruction can achieve the sea-level lowstand observed in many far-field locations at the LGM. Since our model adheres to ice physics, geological observations, and local relative sea-level change, we consider it to be a plausible depiction of the ice sheet configuration at the LGM. Therefore, the origin of the long-debated missing ice problem was likely from the starting assumptions on where ice was distributed and the Earth rheology model, while achieving the far-field sea-level lowstand is a nonunique problem. Second, the ice volume in our reconstruction is unable to match the pre-LGM $\delta^{18}O$ values based on empirical relationships between ice volume (and therefore sea level) and $\delta^{18}O$ but is consistent with some of the sea-level indicators and prior GIA studies[6,9,11,12]. From our results, we propose that these relationships of $\delta^{18}O$ proxy records to sea level and ice volume are not valid.

## Results

**Ice sheet reconstruction overview.** In our approach to ice sheet reconstruction[13,14], we tune the reconstruction in a way that adheres to ice sheet physics, albeit in a very basic way, and obeys constraints on ice margin location and flow direction indicators. Our aim is to avoid using far-field proxy and sea-level records, instead of relying on direct geological evidence of ice sheet evolution and near-field sea-level data. This allows us to assess whether or not there truly is missing ice at the LGM lowstand[15], which happened 26,000–19,000 yr BP (years before present), or if this is an artifact of the assumptions (and circularity) used in creating the previous ice sheet reconstructions. Our method also allows us to independently verify if the low $\delta^{18}O$ values found in deep-sea sediment proxy records prior to the LGM indeed correspond to large ice sheet volume.

Since the GIA signal is dependent on the history of the ice sheet and ocean loading, it is necessary to include a time period prior to our periods of interest, MIS 3 (57,000–29,000 yr BP) and the LGM. For these purposes, we start our reconstruction at 80,000 yr BP, when there was relatively high global sea level[16], and preceding the MIS 4 (71,000–57,000 yr BP) ice advance. Since our reconstruction focuses on general ice sheet configuration changes over the past 80,000 years, we use a relatively coarse time step of 2500 yr BP. This time resolution is insufficient to model near-field Holocene sea-level changes that are sensitive to rapid deglaciation rates but is sufficient for inferring ice sheet configuration prior to the LGM, which has much poorer age control. It is also sufficient to compare with far-field sea-level records that are less sensitive to the exact history of the ice sheets[17].

The global pre-LGM ice sheet extent history was not well developed prior to this study. We, therefore, developed new margin reconstructions for the North American and Antarctic ice sheets based on geological data and incorporated previously published reconstructions for the Eurasian ice sheets (see "Methods"). The main philosophy, especially when reconstructing the North American ice sheets, was to maximize the extent of the ice sheet in order to maximize the possible ice volume. As a result, our North American MIS 3 margin extent is generally larger than the recent assessment by Batchelor et al.[18] (see Supplementary Fig. 8), which was based on an older reconstruction[19]. The validity of the chronological constraints that indicate a reduced ice sheet configuration in North America has been strongly criticized[10], so we present minimal and maximal reconstructions for MIS 3. The minimal reconstruction has a complete retreat of the ice sheet from Hudson Bay for a short period of time in MIS 3 (see Supplementary Fig. 10), while the maximal reconstruction maintains ice cover through all of MIS 3.

**Ice sheet evolution, ice volume, and sea-level change.** For North America, the Laurentide Ice Sheet is initially portrayed as existing only as a small dome in Quebec, which is the likely inception point[20]. The timing of subsequent events is targeted to coincide with the timing of detrital carbonate deposits in the Atlantic Ocean that are correlated with Hudson Strait sourced Heinrich Events[21]. The extent of the ice margin in Hudson Strait is set to reach a maximum at the timing of the event, and subsequent time steps showing a temporally local minimum extent. In the lead-up to the MIS 4 maximum, which we have set to coincide with Heinrich Event 6 (60,000 yr BP), Quebec-centered ice advanced over Hudson Bay, culminating with a merger of the proto-Keewatin dome. The Keewatin Dome was unlikely to have been large prior to the MIS 4 maximum, as the early ice flow directions in southwestern Hudson Bay are oriented directly westward[22]. After Heinrich Event 6, we depict a two-dome structure for the Laurentide Ice Sheet, due to the activation of an ice stream in Hudson Strait, and it is set to remain this way throughout most of MIS 3. In the minimal scenario (Fig. 1a–c), we have set the timing of the retreat of ice from Hudson Bay in MIS 3 to coincide with Heinrich Event 5 (45,000 yr BP), with ice-free conditions only lasting a few thousand years, before becoming ice covered again by 40,000 yr BP. This timing makes it possible to explain the finite-aged radiocarbon dates found in both northern and southern Hudson Bay[5–7]. After this event, there was a gradual expansion of the ice sheet to the culmination of the LGM. In the maximal scenario (Fig. 1d–f), we have set the margin to remain near the MIS 4 maximum limit, except on the east coast, where it is portrayed as remaining near the St. Lawrence River to account for observed margin fluctuations[23–28]. The maximal reconstruction still has margin and basal shear stress changes that are set to

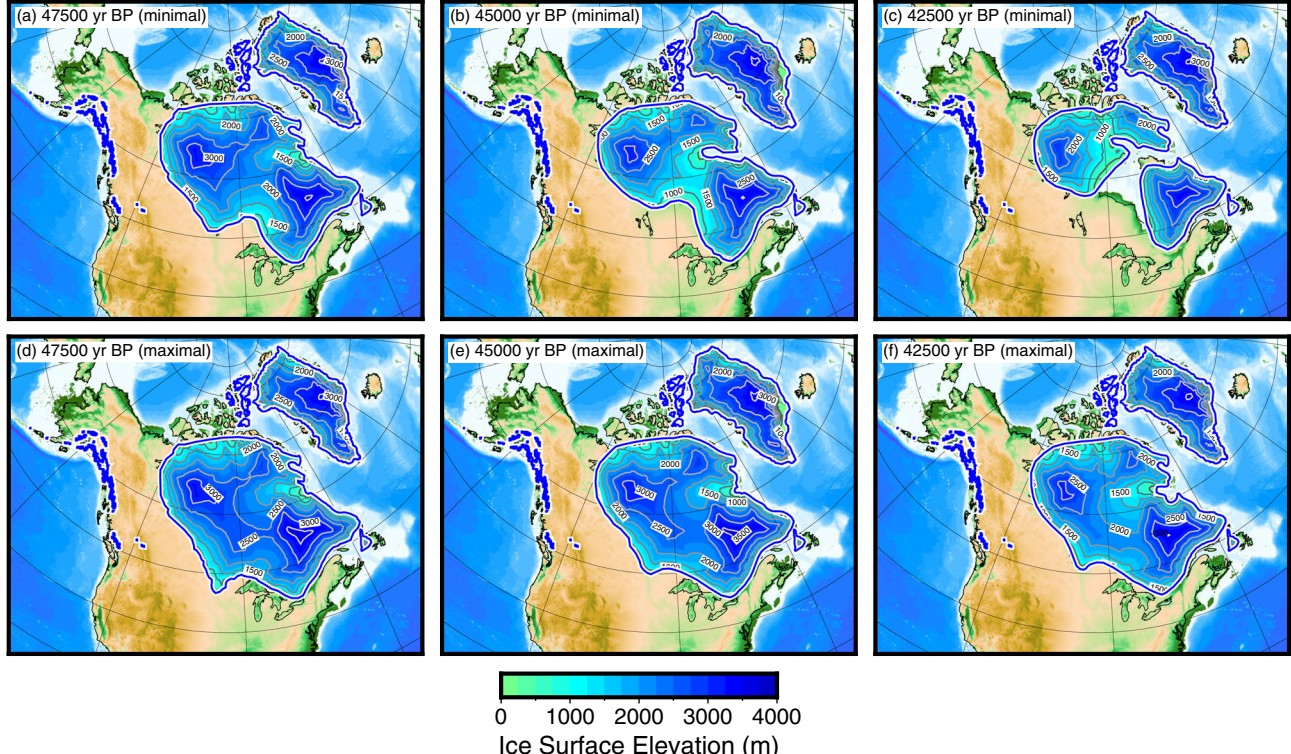

**Fig. 1 Reconstruction of North American ice sheets during Marine Isotope Stage 3. a** 47,500 yr BP—minimal scenario, **b** 45,000 yr BP, prior to Heinrich Event 5—minimal scenario, **c** 42,500 yr BP, after Heinrich Event 5—minimal scenario with full deglaciation of Hudson Bay, **d** 47,500 yr BP—maximal scenario, **e** 45,000 yr BP, prior to Heinrich Event 5—maximal scenario, **f** 42,500 yr BP, after Heinrich Event 5—maximal scenario with less retreat. The contour interval is 500 m.

coincide with Heinrich Event timing. The maximum difference in globally averaged sea level between these two scenarios is about 14 m sea-level equivalent (SLE—the equivalent amount of water, in meters, that would cover the ocean surface if the ice melted) (Fig. 2). Ice volume and SLE for all time steps in the reconstructions can be found in Supplementary Tables 1 and 2.

At present, there are no direct indicators of the size of the ice sheets during the peak MIS 4 glaciation except in northern Europe and in the North American Cordillera (see Supplementary Fig. 11). In our reconstruction, the Eurasian ice sheets have a volume that is comparable to the LGM ice sheets, largely because additional ice in the Kara Sea[29] compensates for the smaller extent further south. The western Laurentide Ice Sheet does not reach the Cordillera, preventing it from achieving volumes similar to the LGM. The resulting global sea-level value for MIS 4 reaches a peak that is ~70% of the LGM volume (Fig. 2).

Changes in the geometry of the Laurentide Ice Sheet dominate the contributions to global sea level in the reconstruction (Fig. 2). Due to this, the predominant tuning strategy was to adjust the Laurentide Ice Sheet reconstruction to ensure that the LGM lowstand was achieved at 20,000 yr BP (years before present) (Fig. 2), while still producing an ice sheet configuration that matched geological evidence for former ice flow patterns[30]. We performed tests using a variety of lower mantle viscosity values, specifically the best-fitting values for far-field indicators[31] ($10^{21}$–$10^{23}$ Pa s). We found the best compromise of maximizing ice volume and near-field sea-level observations[32] was with a high lower mantle viscosity, which we set to $4 \times 10^{22}$ Pa s. The Eurasian ice sheets reconstruction was initially tuned to provide a relatively good fit to postglacial sea-level indicators. Specifically, areas near the edge of the ice sheet[33] required relatively thin ice at the LGM. The GIA response for areas covered by the

Eurasian ice sheets also fit better with our chosen mantle viscosity ($4 \times 10^{22}$ Pa s) than with lower values. Since the GIA response of the Eurasian ice sheets is relatively insensitive to the lower mantle[34], this response is likely influenced by the GIA signal from the North American ice sheets[35]. The Antarctic ice sheets were not tuned to the same extent, as the increase of ice thickness was likely limited at the LGM[36]. The shear stress values on the Antarctic shelf were kept to a low value in order to prevent the excessive ice thickness on continental Antarctica. Ice volume estimates for the Eurasian and Antarctic ice sheets are consistent with estimates from other ice sheet reconstructions and models[15], while the combined Laurentide and Greenland volume is at the lower end of recent estimates.

Since the Eurasian ice sheets were generally restricted to mountainous areas during the middle of MIS 3[37,38], fluctuations in global sea level were controlled almost exclusively by the Laurentide Ice Sheet (Fig. 2). Previous GIA modeling studies[9,11,12] confirmed that the sea level was higher than what was expected from ocean $\delta^{18}$O proxies, due to the reduction of the ice sheet. They predicted sea level as high as about $-40$ m. During much of MIS 3, our calculated sea level is consistent with this value. If Hudson Bay became ice-free as in the minimal scenario, there would have been a sharp rise in sea level, which reached up to $-25$ m. It is unlikely that sea level could have remained as low as $-60$ to $-90$ m as suggested in proxy-based reconstructions[39–41], as even our maximal reconstruction generally remains above $-50$ m between 50,000 and 35,000 yr BP. Few direct sea-level indicators exist during MIS 3 (see Supplementary Figs. 12–17), with some areas (Huon Peninsula, Sunda Shelf) supporting the proxy-based sea-level reconstruction, while other areas (Yellow Sea, eastern United States) supporting reduced ice volume. The reason for the discrepancy in the proxy

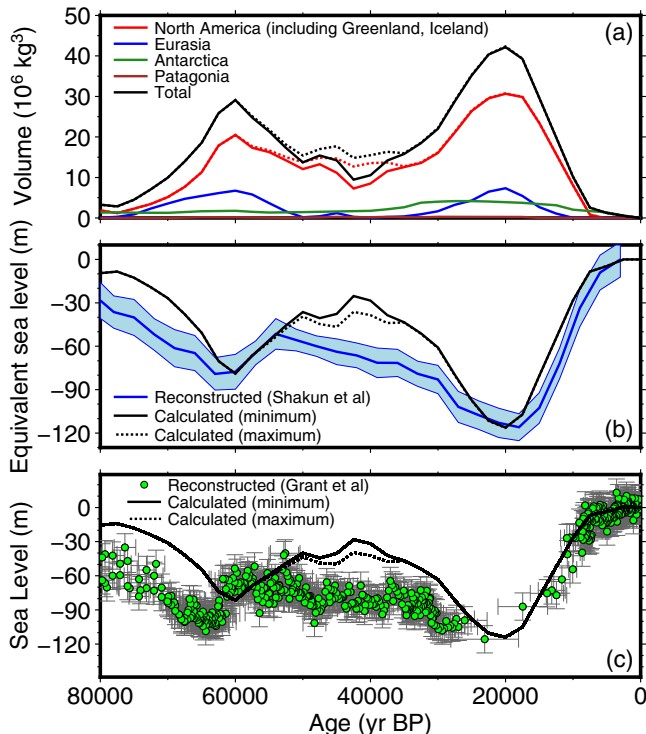

**Fig. 2 Relations between ice sheet volume and sea level in the reconstruction.** The solid line represents the minimal MIS 3 scenario, while the dotted line is the maximal MIS 3 scenario. **a** Ice sheet volume. **b** Calculated ice-volume equivalent sea level, compared with the global seawater $\delta^{18}$O stack[41] (blue line with 2$\sigma$ error range), converted to sea level using a value of $-116$ m for the maximum sea-level drop. **c** Comparison of sea-level calculated at the outlet of the Red Sea and a sea-level reconstruction[40] based on planktonic foraminifer $\delta^{18}$O, with chronology corrected to match an eastern Mediterranean speleothem record (green dots with 2$\sigma$ error bars).

records is unknown. One possible explanation is that the tropical areas where the $\delta^{18}$O records are collected may be more strongly affected by low global temperatures due to relatively low $CO_2$ levels during MIS 3, which compensates for the reduced ice extent in the Northern Hemisphere that is more strongly controlled by insolation. Another possibility is that the duration of the sea-level highstand was too short to have been detected in the proxy records. Given the complexity of interpretation of marine oxygen isotope records, this casts doubts on the reliability of using them to reconstruct past sea level without additional constraints from other lines of evidence (e.g., coral reef sea-level indicators, isotope modeling).

## Discussion

The calculated sea level from our reconstruction is able to match the far-field constraints at Barbados, Sunda Shelf, Bonaparte Gulf, and Great Barrier Reef (Fig. 3). Our maximum calculated ice-volume equivalent sea level (ESL) fall ($-116$ m) and ice volume ($42.2 \times 10^6$ km³) are both substantially less than reported by Lambeck et al.[31] ($-130$ to $-135$ m and $53 \times 10^6$ km³, respectively). We, therefore, find no basis for the missing ice problem[15], as our LGM reconstruction is compatible with existing sea-level constraints, even with a relatively small Antarctic contribution (10 m SLE). Given that our reconstruction does not include components like thermal expansion, groundwater storage, and contributions from smaller glaciers and ice caps (estimated to be 3–4 m SLE[15]), our estimated ice volume is likely slightly

overestimated. The cause of the discrepancy between our results and those of Lambeck et al.[31] probably lies in the difference in the starting ice volume that they used in Antarctica. Lambeck et al.[31] started their analysis with an assumed ice volume of 28 m ESL, which is much greater than the 10 m SLE value in our reconstruction. Due to gravitational effects, more ice will need to be added to Northern Hemisphere to counteract the extra ice in Antarctica to achieve a good fit to far-field sea level, leading to an overall larger ice sheet volume.

An alternative explanation is that the far-field records that show LGM sea-level values below $-120$ m are not representative of the global average sea level. In our model at the LGM (Fig. 3), much of the Southern Hemisphere ocean, as well as coastal regions of southern and southeastern Asia, have calculated sea level below the global average. The global average sea level is only represented in relatively narrow regions, such as in northern South America, western Africa, and Australia (e.g., Bonaparte Gulf and Cairns, Fig. 3). Our model also has somewhat lower (by several meters) sea level in some far-field regions due to our choice of a lower mantle viscosity that is higher than other ice reconstruction studies[42]. In this case, it is not so much that our model fixes the missing ice problem, but that the definition of global sea level is model dependent. We recommend that GIA-based ice sheet reconstruction work should not target a specific ice volume or global sea-level lowstand value, but consider each location as independent observations of relative sea level within a framework of sites around the world.

Our ice sheet reconstruction was created independent of far-field records of sea level. This reconstruction has solved the LGM missing ice problem, and ice volume estimates are consistent with the far-field sea-level drop. Our reconstruction has however created a new missing ice problem, as the $\delta^{18}$O-based sea-level reconstructions during MIS 3 are incompatible with the currently available constraints on ice sheet configuration. Whether this discrepancy can be solved by determining a new relationship between $\delta^{18}$O and sea level, perhaps with the aid of water isotope modeling, or by systematically eliminating current ice sheet extent indicators for MIS 3 should be the focus of future studies. Our reconstruction also demonstrates that there is no consensus on Late Pleistocene ice volume, and we anticipate future refinements, for instance with different Earth rheology assumptions and ice margin histories, will produce different configurations.

## Methods

**Sea-level calculation and GIA modeling.** The GIA was calculated using SELEN, which solves the sea-level equation, including accounting for shoreline migration, adjustments for grounding line position, and rotational feedback[43–45]. This program uses a 1D, spherically symmetric Earth structure. We use a three-layered Earth model with a 120 km elastic lithosphere, an upper mantle with a viscosity of $4 \times 10^{20}$ Pa s, and a lower mantle with a viscosity of $4 \times 10^{22}$ Pa s. This set of Earth model parameters is within the range of values used in Eurasian GIA studies[46], while the lithosphere thickness and upper mantle viscosity are the same as used in the NAICE North America reconstruction[13]. The lower mantle viscosity is compatible with a best-fitting solution in far-field regions[31], but is higher than values used in prior North American GIA-based ice sheet reconstructions[1–3,13,47]. The GIA was calculated to a spherical harmonic degree 256, with three iterations to adjust the ocean function due to changes in the ice load. The sea-level change and Earth deformation were calculated on a 1° grid directly from the output Stokes coefficients to be input back into the ice sheet reconstruction. Since the sea-level calculation was done on a 5 km grid, the 1° grid was projected and interpolated using Delaunay triangulation[48,49]. The development of the reconstruction was done in a progressive manner, where the deformation calculated from a previous ice sheet reconstruction was used to construct an adjusted reconstruction. The final ice sheet reconstruction was iterated twice to ensure that there is close correspondence to the ice load and the calculated sea level and deformation. For a data-model comparison of the sea-level indicators, the sea level was calculated at the location of the indicator, rather than grouping all of the indicators into a region. This is required since there can be large regional variations in calculated sea level, even in far-field locations (i.e., the Sunda Shelf, Fig. 3).

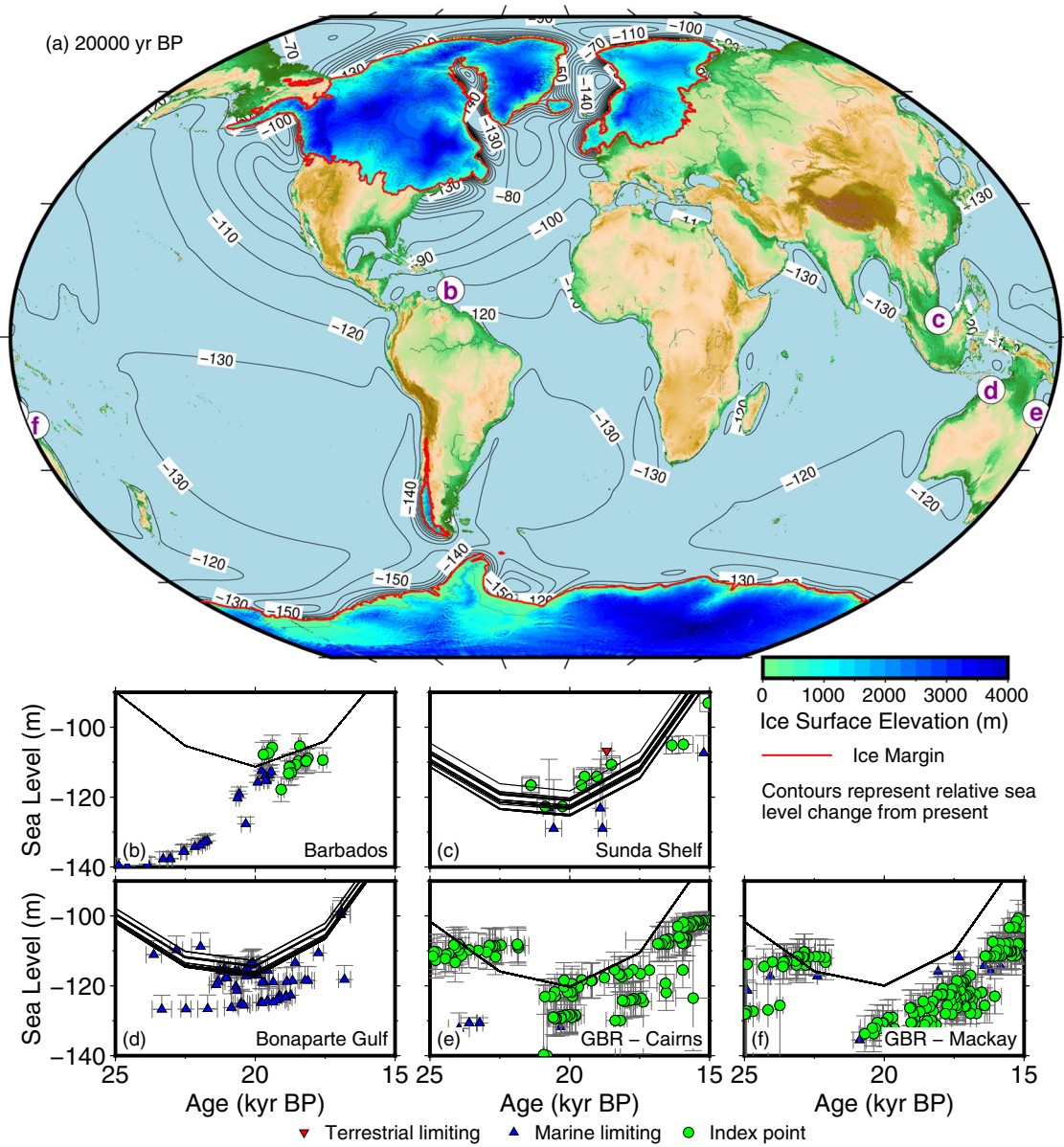

**Fig. 3 Last Glacial Maximum (LGM) ice sheets, paleotopography, and far-field sea level. a** Paleotopography reconstruction at 20,000 yr BP, with contours indicating calculated relative sea level compared to present. Sea level at **b** Barbados[102], **c** Sunda Shelf[103,104], **d** Bonaparte Gulf[105], **e** Great Barrier Reef east of Cairns[106], and **f** Great Barrier Reef east of Mackay[106]. Multiple calculated sea-level curves on the plots (black lines) are due to the fact that the samples come from a broad geographical region, and there is regional variation in relative sea level. We calculate sea level at the location of each sample. The sea-level indicators include index points (sea level is near or at the elevation of the sample), terrestrial limiting (sea level should be below the elevation of the sample), and marine limiting (sea level should be above the elevation of the sample). Error bars represent $2\sigma$ uncertainty in age, and uncertainty in measured elevation and geological context for sea level.

A technical limitation of the version of SELEN we use is that it requires a constant time step, so during the development of the reconstruction, we chose not to pursue more refined time steps during the deglaciation when there is more information, as it would substantially increase the computing time for our exploratory study. This is why this reconstruction has a relatively coarse 2500 year time step. The goal of our reconstruction was to develop the framework to generate global ice sheet reconstructions and then to evaluate general ice sheet change for the past 80,000 years. To test the robustness of the ice sheet reconstruction, we did run a test where we linearly interpolated the ice load to 500 year time steps (see Supplementary Figs. 6–7). In near-field areas, the 500-year time step simulation has a much better fit for the data. In far-field regions where sea level is dominantly affected by ice volume changes, the response is essentially the same. A version of SELEN with variable time steps is in development, and a more refined ice sheet reconstruction is being planned to utilize recently published ice sheet margin reconstructions[50,51].

**Development of ice-sheet reconstruction**. To reconstruct the ice sheets, we use the software ICESHEET[14], which uses an assumption of perfectly plastic, equilibrium conditions. Although the ice sheets were likely never in equilibrium, it produces a glaciologically plausible ice sheet configuration, while requiring only three input parameters, namely margin location, basal shear stress, and basal topography. This makes it suitable for precise control of the ice sheet reconstruction for GIA modeling. The best-constrained part of paleo-ice sheet configuration is the margin, and through our tests, is also the primary control on ice volume. The basal shear stress was adjusted in order to maximize LGM ice volume and reduced downwards during the deglacial period to improve the fit to postglacial sea-level constraints from recent compilations[32,33,52–56]. Due to the 2500 year time steps in this reconstruction, the postglacial sea level in the core areas of the ice sheet are overestimated (since the load does not reduce quickly enough between time steps), so for these areas, the sea-level indicators were only used to give a general sense of how well the model performed. The base topography used in this study is RTopo-2[57]. The ice sheets were calculated on a projected coordinate

system at 5 km resolution. For input into SELEN, the output ice thickness was converted into a hexagonal grid[58] with a resolution of roughly 66 km, which is sufficient for calculating to spherical harmonic degree 256. The workflow is essentially the same as in a previous study[13] but is briefly summarized as follows: (1) First, the ice sheet reconstruction was calculated using modern topography using a shear stress model with values based on surface topography and surficial geology. For Antarctica and Greenland, these values were adjusted to recreate the modern ice sheet surface topography, while for North America and Europe, it was based on the previous studies[13,14,59]. (2) The GIA (Earth deformation and sea level) was calculated and the result was added to modern topography to produce the underlying paleotopography. (3) The ice sheet reconstruction was again calculated using the adjusted paleotopography. This step is necessary to do before analyzing sea level because the deformed topography significantly increases ice sheet volume even with identical shear stress parameters[60]. (4) Calculate the GIA again, and calculate sea level at locations where there are sea-level indicators. (5) Using the comparison of the calculated sea level with the sea-level indicators, we adjusted the basal shear stress up or down, including time dependence (i.e., lowering the value during deglaciation), in an attempt to reduce the misfit, while taking into consideration broad-scale ice flow direction features. (6) Using the previously calculated paleotopography, a new ice sheet reconstruction was determined. Steps 4 and 5 were repeated many times until a generally satisfactory fit was achieved. (7) Since prior to the LGM there are few sea-level indicators, we relied more on matching the inferred changes in ice flow direction through a time when creating the reconstruction. The shear stress and margin locations were adjusted to conform to specific geomorphic and geologic information, which is detailed below. (8) After the targetted geological and sea-level constraints were satisfied, we did a second calculation of the reconstruction without changing the shear stress parameters to ensure that there was a minimal mismatch between the underlying deformed topography, GIA response, and the ice load.

For the flow direction indicators, we assume that the ice sheet flow will be in the direction of the greatest ice sheet surface gradient[61,62]. Thus, by examining geological indicators of ice flow direction, it is possible to evaluate if the ice sheet reconstruction is realistic. Specifically, for the pre-LGM Laurentide Ice Sheet, we made adjustments to the ice margin and basal shear stress to ensure that the ice sheet surface gradient followed known flow direction shifts[22]. While this does not directly constrain ice thickness or volume (and is therefore nonunique), it ensures that for our chosen combination of ice margin and shear stress, that geometry remains internally consistent with geological constraints.

**Margin reconstructions**. The ice sheet margins were developed for the past 80,000 years at the 2500 year time steps. The LGM to present interval utilized existing margin reconstructions[13,36,38,59,63–66]. The margins prior to 20,000 for the Eurasian ice sheets were also based on existing reconstructions[37,38,67]. The main characteristics of the Eurasian ice sheets were that they were limited in extent during MIS 3[37], and not a significant contributor to sea level during this time. Compared to a previous reconstruction[68], the ice extent is less at 40,000–50,000 yr BP, but similar to the minimal extent at 35,000 yr BP. The margins for the pre-LGM period for Antarctica and North America were drawn from scratch for this study, details which are expanded on below. We have also included the Patagonian Ice Sheet[66], but it contributes less than 1 m SLE ice volume, even at the LGM. Chronology of Patagonian Ice Sheet fluctuations is based on dated landforms in southern Patagonia[69]. For time slices that are intermediate of the timing of existing margin reconstructions, we set the margin to be intermediate of them.

The pre-LGM constraints on the Antarctic ice sheet are sparse. In parts of the West Antarctic Ice Sheet, the maximum extent at the continental shelf edge may have been reached prior to the LGM[70]. This was likely attained about 30,000 yr BP, which is the time slice we set the maximum extent. In the Ross Sea sector, the margin may have gradually advanced to the shelf edge from some minimum limit in MIS 5e[71]. This is used as a template for the entire West Antarctic Ice Sheet, which we depict as gradually advancing from an extent that was similar to present at 80,000 yr BP, to a maximum extent at 30,000 yr BP. The East Antarctic Ice Sheet remained close to its present-day extent through MIS 3[72,73], so we have used the present-day margin throughout MIS 4 and 3.

Fluctuations in sea level for the past 80,000 years are dominated by changes in the Laurentide Ice Sheet, so it is most sensitive to ice sheet margin location uncertainties. Since the error ranges of many of the chronological constraints become large the further back in time they go, the chronology of advance and retreat events was set to coincide with the timing of Heinrich Events[21]. For this reconstruction, the timing of the Heinrich Event represents a local maximum ice-sheet configuration, and the subsequent time step shows a retreat from that position. The 2500 year time step used in our reconstruction fits conveniently with the roughly 7000-year intervals of Heinrich Events. It must be acknowledged that the margin chronology we propose has uncertainty on the order of several thousand years. We utilized the recent compilation of pre-LGM indicators of ice-free conditions[6] to guide the ice extent history for MIS 3. In the minimal scenario, the data that indicate ice-free conditions in Hudson Bay existed are used, while in the maximal scenario, these are ignored. The calculated Holocene sea level within Hudson Bay is not sensitive to the choice of scenario. The maximum extent of the MIS 4 glaciation is set to be 60,000 yr BP, which coincides with Heinrich Event 6, after which there is a retreat of the ice sheet. Retreat of the margin is also set to

happen at 50,000, 42,500, and 37,500 yr BP, which are after Heinrich Events 5a, 5, and 4. The MIS 3 deglaciation of Hudson Bay in the minimal scenario is set to be at Heinrich Event 5 rather than 5a, due to the fact that the radiocarbon ages are finite (which would not be the case during 5a), and that this event was likely short lived enough that it did not produce recognizable erosional features in far-field regions. In the maximal scenario, the ice sheet south of Hudson Bay is set to remain in northern Ontario and southern Manitoba, with fluctuations that are coincident with the Heinrich Events. In this scenario, it is assumed that the presence of the Laurentide Ice Sheet saddle between the Keewatin and Quebec–Labrador centers remained near the southern end of Hudson Bay throughout MIS 3, which may explain why the pre-MIS 2 sediments were preserved. The western margin of the Laurentide Ice Sheet is set to remain in the vicinity of Great Slave Lake during MIS 3, based on recent work that presents evidence of ice-free conditions west of the lake prior to the LGM[74]. The Laurentide Ice Sheet did not reach Banks Island during MIS 4 and 3[75], so the limit is set to be east of there on Victoria Island. The ice sheet did not advance over Lake Superior and into Lake Michigan until near the end of MIS 3[76], and so this is set to be the southern limit during MIS 4. The ice margin advanced and retreated into southern Ontario and the St. Lawrence Valley several times between 80,000 and 30,000 yr BP. This reflects the relative stability of the Quebec–Labrador dome of the ice sheet. Our margin reconstructions in this region are based on records of these fluctuations[23–28]. The MIS 4 extent in the Eastern Canada region is set to reach the shelf edge off the coast of Nova Scotia[77]. There is some evidence of a possibly brief retreat of the ice sheet in eastern Baffin Island during MIS 3[78], but we have kept the margin near the edge of Baffin Island throughout MIS 3 and 4. The evolution of the margins in the central parts of the ice sheet was based on the relative chronology of major changes in flow direction[20,22,79–81]. The margin reconstructions between 30,000 and 20,000 yr BP are described in detail in Gowan[60].

The Cordilleran Ice Sheet history is defined as having major glacial episodes during MIS 4 and MIS 2, with little ice cover during MIS 3. The Late MIS 3 margins are based on the previous reconstructions[82]. Prior to the beginning of advance after about 35,000 yr BP, the ice extent may not have been much different than the present, and non-glacial sediments are preserved throughout British Columbia[83–85]. In many areas of the Cordillera, the MIS 4 maximum extent was the same or even somewhat larger than the LGM limit[86–91]. The MIS 4 glacial maximum for the Cordilleran Ice Sheet happened at about 55,000 yr BP[86,88], which is somewhat later than what we set the Laurentide Ice Sheet (which was based on the timing of Heinrich Event 6[21]).

The Greenland Ice Sheet in MIS 4 was similar in extent as the MIS 2 maximum[92–94]. The retreat during MIS 3 was in some places substantial, such as in northeastern Greenland, where the extent is set to be less than present[95]. The northwestern Greenland Ice sheet also may see a similar retreat during MIS 3[96]. The Holocene margin is based partially on the modern-day extent[57], with adjustments in western Greenland to take into account the Holocene thermal maximum[94].

For the Innuitian Ice Sheet in northern Canada, evidence for pre-LGM late Pleistocene glaciation beyond the modern ice cap limits is lacking[97,98]. The buildup of this ice sheet may rely on the Greenland Ice Sheet advancing over the Nares Strait, which may not have happened prior to the LGM during the Late Pleistocene[99]. As a result, we have kept the modern extent ice caps as the margin limit for the pre-LGM period.

**Adjustment of ice-sheet reconstruction**. After finalizing the margins, the basal shear stress was adjusted to increase or decrease the thickness of the ice sheet for modeling sea level and to match large scale indicators of ice flow direction. By increasing the basal shear stress, the ice sheet surface slope is steeper and results in thicker ice. As mentioned above, the main goal was to be able to maximize global ice volume at the LGM, while balancing near-field Holocene data. The initial tuning was purely to achieve the originally inferred global 130-m sea-level drop by increasing the thickness of the core of the Laurentide and Eurasian ice sheets, but achieving this produced unrealistically thick ice sheets (i.e., over 5-km thick in places). The first sea-level datasets added to tune the basal shear stress, and therefore ice thickness, were within the core regions of the ice sheets, and it became necessary to reduce the shear stress from the initial values that we used.

Although in most places there are few physical constraints on the thickness of the Northern Hemisphere ice sheets at the LGM, there are some general patterns that can be deduced by looking at the pattern of Holocene sea-level change. First, mountainous areas have much higher shear stress than areas with relatively flat terrain. This was directly observed when we tuned the shear stress values for the Greenland[14] and Antarctic ice sheets to reproduce the modern ice-sheet configuration. Secondary control is that areas with continuous sediment cover have lower shear stress than places with discontinuous sediment cover[100]. This can be seen in the fact that the peripheral regions of the ice sheets were relatively thinner than the core areas to accommodate pro-glacial lake strandline tilts[13]. The third control on the shear stress was that regions with major ice streams have lower shear stress than surrounding areas. This is particularly true for the Hudson Strait, where the shear stress had to be significantly lower than the adjacent Quebec–Labrador region in order to produce a much thinner ice load within Hudson Bay. During deglaciation, the shear stress values generally needed to be reduced in order to get closer correspondence to the GIA response. This lowering of shear stress is likely a

result of increased basal meltwater and warming of the base during deglaciation and increased interactions with pro-glacial lakes and the ocean at the margins. Since topographic variability is the main factor controlling the shear stress, the geographical domains were divided on this basis, to separate mountainous areas from areas with flatter topography. After this step, the regions were further adjusted to account for variability in surficial geology. After some initial tuning, we tested the impact of varying the lower mantle viscosity on postglacial sea level in Hudson Bay. The initial Earth model that was utilized had a viscosity of $4 \times 10^{21}$ Pa s, but the modeled sea level was much higher than the observations. Tests using different lower mantle viscosity values revealed that values approaching $10^{23}$ Pa s reduced the postglacial sea level. Using a higher viscosity value made it possible to have a larger ice load in the center of the Laurentide Ice Sheet, and therefore maximize ice volume, so a value of $4 \times 10^{22}$ Pa s was chosen for subsequent tuning.

Compared with our earlier reconstruction[13], which was only tuned on the basis of GIA constraints in the western Laurentide Ice Sheet and using a lower value for lower mantle viscosity, the core region (Hudson Bay and the Quebec–Labrador sector) of the ice sheet is ~1000–1200 m thicker, while some of the peripheral regions, such as the Great Lakes, northeastern United States, and Boothia Peninsula are thinner by 300–500 m. The general two-dome structure with lower elevation within Hudson Bay remains.

**Evaluation**. We have created two reports comparing calculated sea level to sea-level indicators from a global database[42]. We have included the plots from Churchill (located near the center of the Laurentide Ice Sheet) and Ångermanland (located near the center of the Eurasian ice sheets) in Supplementary Figs. 2–7. One report shows the results using six different Earth models using lower mantle viscosity values between $10^{21}$ and $10^{23}$ Pa s. This demonstrates that using a lower mantle viscosity toward $10^{23}$ Pa s improves the fit of the calculated sea level in ice-covered areas. The second report shows the results from the standard version of Paleo-MIST, a 500-year time-step version where the ice load was linearly interpolated, the variant with Hudson Bay remaining ice-covered through MIS 3, plus three commonly used Earth models[1,2,101]. The high temporal resolution version provides a better fit to sea-level indicators in formerly glaciated areas, with lower sea level since the ice load is reduced in a more gradual, realistic way. The available sea-level indicators during MIS 3 cannot distinguish which Laurentide Ice Sheet scenario is more likely (see Supplementary Figs. 12–18). There is also almost no sensitivity to the scenarios on deglacial sea level.

## Data availability
PaleoMIST 1.0, which includes the ice sheet margin, paleotopography, and ice thickness datasets and Stokes coefficients produced from this study are available on Pangaea (https://doi.org/10.1594/PANGAEA.905800). Reports showing this evaluation, plus the calculated sea level can be found at the following link: https://doi.org/10.5281/zenodo.4061594.

## Code availability
ICESHEET 2.0, which includes the scripts necessary to reproduce the reconstructions in this study, can be found on Github, (https://github.com/evangowan/icesheet) and as an archive on Zenodo (https://doi.org/10.5281/zenodo.4311508). Scripts to compare the calculated sea level and sea-level indicators can also be found on Github (https://github.com/evangowan/paleo_sea_level). Reports using these scripts, plus the calculated sea level can be found on Zenodo: (https://doi.org/10.5281/zenodo.4061594).

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

## Acknowledgements

We acknowledge Holger Steffen for helpful discussions on the ideal Earth model parameters and for access to an unpublished database sea-level indicators in the Baltic Sea region. E.J.G. was funded by Impuls- und Vernetzungsfonds, Helmholtz-Exzellenznetzwerke (grant no. ExNet-0001-Phase 2-3) "The Polar System and its Effects on the Ocean Floor (POSY)" and Helmholtz Climate Initiative REKLIM (Regional Climate Change), a joint research project at the Helmholtz Association of German research centers (HGF). Further support to this work has been provided by the project SEASCHANGE (RO-5245/1-1) from the Deutsche Forschungsgemeinschaft (DFG). This study was also supported by the PACES-II program at the Alfred Wegener Institute and the Bundesministerium für Bildung und Forschung funded project, PalMod. X.Z. was funded by National Science Foundation of China (42075047), National Key R&D programme of China (2018YFA0606403) and Natural Science Foundation of China-Shandong Joint Fund for Marine Science Research Centers (U1606401). We acknowledge PALSEA, a working group of the International Union for Quaternary Sciences (INQUA) and Past Global Changes (PAGES), which in turn received support from the Swiss Academy of Sciences and the Chinese Academy of Sciences for constructive discussions during the Dublin meeting (2019). Figures in this paper were plotted with the aid of Generic Mapping Tools[49].

## Author contributions

E.J.G. led this study, including the development of the ice sheet reconstruction software, construction of the pre-LGM margins, and analysis of GIA modeling. X.Z. led the interpretation of the climatic implications of the reconstructions, with input from G.L. S.K. assisted E.J.G. in developing the scripts for ICESHEET and completed the reconstruction of northern Canada. A.R. contributed to the interpretation of sea-level proxy data. P.S. developed the version SELEN used in this study. A.H., R.G., J.M., and J.I.S. created the margin reconstructions for Eurasia from mid-MIS 3 to present and provided data and feedback related to the Eurasian reconstruction. All authors contributed to the study design and writing of the paper.

## Funding

## Competing interests

The authors declare no competing interests.
