## [Peer Review File · Nature Communications]

Reviewer comments, first round –

Reviewer #1 (Remarks to the Author):

The manuscript "A new global ice sheet reconstruction for the past 80,000 years" presents the first ice sheet reconstruction for the last 80 ka that is independent of far-field records. A simple ice sheet model constrained by geological observations are used to reconstruct ice sheet history and volume only by near field sea level indicators. The new reconstructions (PaleoMIST 1.0) enable the authors to address a very controversial topic in paleoclimate science; the "missing ice" during LGM where there is a large discrepancy (8-28 m) between estimates of ice volume based on ice sheet reconstructions and sea level records from far-field records. In addition, they estimate a minimum and maximum MIS 3 ice extent, and calculate ice volume and find that both scenarios are significantly smaller than far-field data is suggesting. They claim that there LGM reconstruction is consistent with far-field data and furthermore highlight that $\delta^{18}O$ is not fully understood and one should be caution to use the record as ice volume indicator. In principal, this seems to be a novel approach and scientific results that potentially could be suitable for publication in Nature Communications.

However, I have some concerns about the study. The overall premise is that there is a mismatch between near-field geological information about ice sheet size and far-field sea level indicators. The authors furthermore write that most studies use far-field sea level data to tune ice sheet models which results in very large ice sheets to account for the sea level lowering. In the new study the authors only use near-field data and a simple ice sheet model to estimate ice volume and sea level lowering during LGM without tuning the model using the far-field data. Their max. model suggests 20% less ice volume than other tunes models, but at the same time they claim that is consistent with far-field data. How can that add-up? The starting premise was that far-field data might be erroneous and should not be used to tune models – how can it suddenly be consistent? Apparently, their data is consistent with far-field data from Barbados, Sunda Shelf and Great Barrier Reef. But on the other hand, it is inconsistent with Lambeck et al (2014) which is based on the same far-field data but corrected for isostatic effects and it indicate +130 m lower sea level at LGM. Is the GIA correction the problem? If so, it should be elaborated.

In summary, although the study is intriguing and potentially suitable for Nature Comm., the presentation of the results and discussion is not clear and there are inconsistencies that raises concern about the overall outcome to the study. If this can be accommodated the manuscript has the potential to make a big impact

Reviewer #2 (Remarks to the Author):

Review of manuscript: reference no: NCOMMS-20-08917-T: "A new global ice sheet reconstruction for the past 80,000 years".

The authors have presented a very novel approach to create an entirely new global ice sheet reconstruction, without relying on previously published global models. This is not an easy task. They begin to address many different issues in the manuscript - MIS3 North American ice sheet reconstruction, LGM sea level budget, comparison to $\delta^{18}O$ and far-field sea level data. However, for publication in Nature Communications would require more work to fully support the methods and many statements/conclusions given in the paper. Without these I would not support publication. However, if the paper is revised it would make a value publication for within the journal.

I have separated my comments into a series of sub sections with a theme and highlighted in italics the relevant parts of the manuscripts - with page number (line number in brackets).

Abstract (and title): 'a global ice sheet reconstruction for past 80ka'; Page 2(21) 'we present a global ice sheet reconstruction that is independent of far-field records for the past 80,000 ka'. The manuscript does not present a model from 80ka to present day. The entire interval is given on Fig.2; but comparison to far-field sea level is up to 15 ka (Fig.3) and it is stated Page 3(57) 'this time resolution is insufficient to model near-field Holocene sea-level changes'. How do you justify the presented model from 15 ka to present? If the model is not viable to model near-field sea level (due to time resolution); have you produced a simulation that is not suitable to be used for 30 ka to present day near-field sea level modelling?

Sea level model: You do not use a version of the code which includes with rotational feedback to treatment of retreating marine-based ice (Page 13 (169) . These are both important processes that should be included for sea-level modelling. Can you comment on this? Page 143 (178): 'The GIA was calculated to a spherical harmonic degree of 256...' 'The sea-level change and earth deformation was calculated on a 1 degree grid'. If the GIA model is run at 256 (~ 70 km); how do you convert to a 1 degree (~ 110km)? This would involve some software for remodelling? projection? Is 1deg this the resolution of ICESHEET used when producing the input ice sheet reconstruction?

Earth model: Page13(173): 'This set of Earth model parameters is within the range of values used in Eurasian GIA studies'. Page 6(121) 'the GIA response in areas covered by the Eurasian ice sheet are not sensitive to changes in the lower mantle viscosity'. Why do you adopt the reference of values from the Eurasian ice sheet for the lower mantle, if you have determined (no evidence presented) it is not sensitivity to constrain this parameter?

The main focus of the paper is on the sensitivity changes in the North American ice sheet, does it not follow to use an earth model preferred by the North American ice sheet (or GIA-based studies of this ice sheet). Steffan and Wu, 2011 the average value for lower mantle (Table3 in paper) is closer to ~ 2×10^{22} Pas.

Page6 (118) 'we perform tests using a variety of lower mantle viscosity values'. What range of models? what tests? - Did you run all simulations for a suite of models? you state you use models that are from a far-field study. However, as above, your model development is based on near-field evidence only, so why adopt an earth model from far-field reconstruction?

Time step: Page 3(56) 'Since our ice sheet reconstruction focuses on general ice sheet configuration changes over the past 80 ka, we use a relevantly coarse time step of 2.5 ka...it is insufficient to model..Holocene... rapid sea-level rates'. There is a rapid change in the RSL rate ~ 20 ka with the inflection at the onset of the deglaciation. Is 2.5 ka sufficient to capture this? Page2(23) 'thickness is controlled by near-field sea level indicators'. Page 14 (194) 'Due to the 2.5ka time step in this reconstruction, the post-glacial sea level in the core areas of the ice sheet is overestimated'. Page 19 (296) 'the basal shear stress was adjusted ... to match the ice flow direction'. Initially it is stated that near-field sea level is used to control thickness, but the time step is too coarse to model in the core areas. However, in the full method description you state the flow lines are used? Please be clear. I understand and it is approach to use the coarse resolution when there is little data, but how have you evaluated the impact on the modelling? why not reduce later in the reconstruction?

LGM sea level budget. The authors have presented one reconstruction which captures the upper error limits of the far-field sea level data (Fig.3). This is great and not an easy task. However, to fully evaluate this, also requires evaluation of the period post LGM. From Fig3, the authors only evaluate the model to the far-field sea level up to ~ 15 ka; why disregard the entire record?. The model underpredicts the delta 180 over the deglaciation (Fig.2) and GBR data. Page 7 (136): 'Given our reconstruction does not include components.... the ice volumes are likely overestimated' From this statement, does it imply that your results would be ~ 3-4m lower, increasing this underprediction?

Near-field landform data/Ice sheet reconstruction: The approach of developing a complete new ice sheet reconstruction from scratch (for want of a better phrase) is a complicated task and the work the authors present is good. The development of a new approach to do this, given the issues

(circularity) with using far-field data is great. However, there are a number of very general statements (both in the main paper and Method) of how ice sheet reconstruction is evaluated or developed in comparison to near field data for which evidence is not provided. The method describes the use of ice flow direction: How is this related? do you model ice streams? a GIA model does not have ice flow (or the plastic ice sheet model).

The Eurasian ice sheet reconstruction which incorporate the DATED-1 information, is great. The constraints in paper DATED-1 as the authors allude to, do not extent into MIS4. However, Lambeck2010 'The Scandinavian ice sheet: MIS4 to the end of the LGM' reproduce an reconstruction, using evidence from earlier MIS4. How does your reconstruction compare during this earlier period.

Near-field sea level data: In the SOM the modelled RSL is compared to a very limited number of Holocene sea level data? There is no reference to these results in the paper. What near-field sea level data was used in the development of the North American ice sheet. If it is only the MIS4/3 data, how is the 30 ka to present day evaluated (without using the far-field data). I would consider it necessary to evaluate (even using some simple statistics) the fit of the rsl predictions to your LGM far-field sea level data.

Reviewer #3 (Remarks to the Author):

The manuscript 'A new global ice sheet reconstruction for the past 80 000 years' by Gowan et al. is a much needed study that attempts to shed light on the evolution of the ice sheets of the last glaciation from a global perspective. The study is based on glacial-isostatic adjustment (GIA) methods and I assume the other approached reviewers have full insight into that, which I don't. With my main expertise in palaeoglaciological reconstructions based on empirical data, I will focus my review on the empirical information on ice sheet margins that the authors use as one of the main inputs into their reconstruction.

The authors utilise previously published information for the Eurasian ice sheets and carry out their own literature survey to source information for the remaining ice masses (described at lines 221-294). I generally find their described approach sound and well-informed. One point that I would raise is that given the information availability, some better documented minor ice masses, such as the Patagonian Ice Sheet or the Brooks Range, got included, while the glaciation of Siberia, that at MIS 4 likely contained a sea level equivalent of more than 1 m, was not included. But I understand that this is likely because of the lack of information for that region.

For the Laurentide Ice Sheet, the history of which is also poorly understood before the LGM, the approach of using the Heinrich events as a proxy for the position and timing of the eastern margin is probably sound, even though it hinges the history of the ice sheet on a set of events that we still understand rather poorly.

The authors refer briefly to the recent publication of Batchelor et al. (2019) that aimed to compile the available information on the extents of Northern Hemispheric ice sheets at selected time slices / periods prior to the LGM (the paper contains 6 time slices relevant for this study). It would be interesting to see even a brief discussion of matches and mismatches between the outlines of Batchelor et al. and those of this reconstruction.

I hope that the authors plan to make the modelled results available in a way that other GIA reconstructions, such as that of Lambeck et al. (2017) or Peltier et al. (2015), have been. A short section on data availability could be added for that.

To conclude, I believe that this will be a widely used resource for many in the Quaternary / palaeoclimate community. The manuscript is well structured and written in a good style. I recommend it for publication in Nature Communications.

I hope that the authors will find my comments useful.

Best regards,

Martin Margold, 30 March 2020

Minor comments:

L 241 the error ranges become

L 243 the chronology was set

L 282-283 'The MIS 4 glacial maximum for the Cordilleran Ice Sheet happened at about 55 000 yr BP, which is somewhat later than what we set the Laurentide Ice Sheet'. This statement needs to be either referenced or briefly explained.

A new global ice sheet reconstruction for the past 80 000 years – response to comments

Evan J. Gowan^{1,2}, Xu Zhang^{1,3}, Sara Khosravi⁴, Alessio Rovere², Paolo Stocchi⁵,
Anna L. C. Hughes^{6,7}, Richard Gyllencreutz⁸, Jan Mangerud⁷, John-Inge Svendsen⁷
& Gerrit Lohmann^{1,2}

October 9, 2020

1. Alfred Wegener Institute, Helmholtz Center for Polar and Marine Research, Bremerhaven, Germany
2. MARUM, University of Bremen, Bremen, Germany
3. Key Laboratory of Western China's Environmental Systems (Ministry of Education), College of Earth and Environmental Science, Centre for Pan Third Pole Environment (Pan-TPE), Langzhou University, Langzhou, China
4. Alfred Wegener Institute, Helmholtz Center for Polar and Marine Research, Potsdam, Germany
5. NIOZ, Texel, Netherlands
6. Department of Geography, University of Manchester, Manchester, United Kingdom
7. Department of Earth Science, University of Bergen and Bjerknes Centre for Climate Research, Bergen, Norway
8. Department of Geological Sciences, Stockholm University, Stockholm, Sweden

1 Overview

First off, we would like to thank the three reviewers for their comments. All three reviewers were supportive of our work, though reviewers #1 and #2 had concerns that were mainly focused on the need for backing up some of the claims and statements we made. This is entirely reasonable, and in our revised manuscript and supplementary analysis, we hope that we have provided enough information to back up our claims. Here, we summarize the main steps we have taken to address the comments. The original comments from the reviewers are in italics, our response is in standard font, and changes to the text in the manuscript are shown in blue. Note that all of the reconstructions and code are available for download, as noted in the data availability section (which may have been overlooked originally, since it fell behind the references for some reason). Our datasets have been uploaded to Pangaea, and will be available to anyone (<https://doi.pangaea.de/10.1594/PANGAEA.905800>).

1.1 Comparison of sea level data and the modelled sea level

Several comments mention that they would like to see a more thorough comparison of all of the sea level indicators we used for the development of the reconstruction, especially for the whole deglacial period. Though we compared our model with sea level indicator data, using a metric developed by Gowan et al[1], this was initially not done in a way that made it easy to compare different model results. As a result, we have created a series of scripts to create a report that is able to summarize the relative fit of six different ice/Earth models in a single document. These scripts can be found on Github (https://github.com/evangowan/paleo_sea_level). The references to all the data are included in these reports.

The sea level indicators used for our reconstruction includes regions covered by Northern Hemisphere ice sheets (Canada and northern Europe), the Siberian coast, Eastern United States, southeastern Asia, and selected far field locations (northern Australia, Barbados, Tahiti). Though this is not global coverage (it is a considerable amount of work to assemble the datasets for analysis), we believe it provides enough constraints to assess both the size and geometry of the Laurentide and Eurasian ice sheets, as well as assess the total ice volume of all ice sheets. The generated reports, plus the calculated sea level used in the reports, have been archived[2] (<http://doi.org/10.5281/zenodo.4061594>).

Two reports have been generated, which will be elaborated on below. One report shows the results of six different lower mantle viscosity models (with the ice model and other Earth model parameters kept constant. The second report shows the results of the standard PaleoMIST 1.0 reconstruction, including the models with and without deglaciation of Hudson Bay during MIS 3, a high temporal (500 year) resolution model, and our reconstruction with three Earth models based on those favoured by James et al[3] (for tectonically active areas), Peltier et al[4] and Lambeck et al[5].

We have added a subsection in the methods section to give a brief overview.

We have created two reports comparing calculated sea level to sea level indicators from a global database[2]. We have included the plots from Churchill (located near the center of the Laurentide Ice Sheet) and Ångermanland (located near the center of the Eurasian ice sheets) in supplementary figures 2–7. One report shows the results using six different Earth models using lower mantle viscosity values between 10^{21} and 10^{23} Pa s. This demonstrates that using a lower mantle viscosity towards 10^{23} Pa s improves the fit of the calculated sea level in ice covered areas. The second report shows the results from the standard version of PaleoMIST, a 500 year time step version where the ice load was linearly interpolated, the variant with Hudson Bay remaining ice covered through MIS 3, plus three commonly used Earth models[3, 4, 5]. The high temporal resolution version provides a better fit to sea level indicators in formerly glaciated areas, with lower sea level since the ice load is reduced in a more gradual, realistic way. The available sea level indicators during MIS 3 cannot distinguish which Laurentide Ice Sheet scenario is more likely. There is also almost no sensitivity to the scenarios on deglacial sea level.

1.2 Choice of lower mantle viscosity

The choice of using a relatively high lower mantle viscosity (4×10^{22} Pa s) came after discussions with Holger Steffan, who looked at the optimal choice when modelling Fennoscandia. During the initial phases of creating our reconstruction, we found that using this value also made it easier to add more ice to the Hudson Bay region compared to a lower value that has been traditionally used in GIA based ice sheet reconstructions. Originally we had made one plot showing a comparison of different lower mantle viscosity values in Churchill (figure 2 in the supplementary material), but now we show in our report a comparison with all of the sea level data we utilized. We have calculated the sea level response using six different lower mantle viscosity values in our first report. The viscosity values range between 10^{21} and 10^{23} Pa s.

The results of this report back up our claim that our choice of lower mantle viscosity (4×10^{22} Pa s) provides a better fit to the sea level data in glaciated areas. Although our initial claim was that the Eurasian ice sheet was not sensitive to the choice of viscosity (since the ice sheet is smaller in extent), the more thorough analysis also reveals that a better fit is achieved using our selected mantle viscosity.

One caveat with all of this is that our reconstruction was tuned to our chosen viscosity value of 4×10^{22} Pa s, so it does not necessarily prove that this value is true. Future studies should be repeated using different Earth rheology values and see if it is possible to create a different ice sheet histories that are consistent with the sea level data.

To provide a demonstration of our claim, we have added figures in the supplementary materials (figures 2–5 in the revised supplement) comparing the calculated sea level in two locations near the center of the ice sheet complexes in North America and Europe, specifically at Churchill (on the coast of Hudson

Bay) and Ångermanland (On the coast of the Baltic Sea). The figures are reproduced in the following two pages.

Figure 1: Map showing the location of sea level indicators in the Churchill region of southern Hudson Bay[6].

Figure 2: Sea level indicators in the Churchill region of southern Hudson Bay[6] and calculated sea level at the location of each indicator (the spread in the response is due to spatial variations in calculated sea level). The calculated sea level is lower using a higher lower mantle viscosity and therefore closer to the sea level inferred from the sea level indicators. Using a higher lower mantle viscosity therefore makes it easier to simultaneously include larger ice volumes in central Canada and fit the sea level indicators. Note that the calculated sea level is still too high, which is at least partly a consequence of the low temporal resolution of the reconstruction and the fact that SELEN treats the load between time steps as a heaviside function. This will cause the load to be overestimated during deglaciation, a factor that will be mitigated in future studies when a higher temporal resolution will be used.

Figure 3: Map showing the location of sea level indicators in the Ångermanland in the Baltic Sea (data from an upcoming compilation by Rosentau *et al.*)

Figure 4: Sea level indicators in the Ångermanland in the Baltic Sea (data from an upcoming compilation by Rosentau *et al.*) and calculated sea level at the location of each indicator (the spread in the response is due to spatial variations in calculated sea level). The calculated sea level is lower using a higher lower mantle viscosity and therefore closer to the sea level inferred from the sea level indicators. Although it is not expected that the lower mantle is sensitive to the Eurasian Ice Sheet, the response is improved by using the higher lower mantle viscosity. The response may be affected by the North American ice sheets[7]. Note that the calculated sea level is still too high, which is at least partly a consequence of the low temporal resolution of the reconstruction and the fact that SELEN treats the load between time steps as a heaviside function. This will cause the load to be overestimated during deglaciation, a factor that will be mitigated in future studies when a higher temporal resolution will be used.

1.3 Time resolution

The reviewers expressed concern that the low temporal resolution (2500 years) used in the reconstruction may lead to incorrect results, and that this should be tested. The choice of this kind of temporal resolution was due to a technical limitation of SELEN, as it required constant time steps. As we were interested in general ice sheet evolution for a longer period of time, this interval was chosen for practical considerations (since the uncertainties on ice sheet margin chronology becomes large before about 30 000 years BP, and using higher resolution makes little sense). Using a higher temporal resolution that is required for evaluating deglacial sea level substantially increases the computing time. We originally stated that since SELEN uses a Heaviside function for computing the load, that it would overestimate sea level in areas that were glaciated.

To test that, we have run a simulation using a 500 year time step, by linearly interpolating the ice load between the time steps. We used the ice loading history based on the scenario where Hudson Bay was ice free during MIS 3. The results, which are shown in the second report, support our assertion that the sea level in glaciated area was overestimated in the original low temporal resolution case. In almost all areas, the higher temporal resolution provides a better fit to deglacial sea level, and provides support that the ice geometry in our reconstruction is appropriate. In far field areas, the the difference between the low and high temporal resolution results is negligible. This is not unexpected, since these areas will be mostly sensitive to changes in global ice/ocean volume. There is a slight difference between the two simulations around 2500 yr BP, as the low resolution simulation doesn't capture the approximately 1.7 m rise in sea level between 2500 and present (due mostly to retreat of Antarctic ice shelf grounding lines). This discrepancy does not substantially impact the scoring metric we use (which is the sum, in meters, of the discrepancy between the calculated sea level and sea level data points for the entire region), which is not strongly influenced by late Holocene sea level.

In addition to showing the calculated sea level at all sites in the report, we have also included plots of the comparison in figures 6 and 7 of the supplementary material for Churchill and Ångermanland. We have included these figures below:

Figure 5: Sea level indicators in the Churchill region of southern Hudson Bay[6] and calculated sea level at the location of each indicator (the spread in the response is due to spatial variations in calculated sea level). This plot shows the difference between the standard version of PaleoMIST 1.0 with 2500 year time steps, and a model run where the load is linearly interpolated to 500 year time steps. The higher temporal resolution better captures the deglaciation, and the calculated sea level is closer to the observations.

Figure 6: Sea level indicators in the Ångermanland in the Baltic Sea (data from an upcoming compilation by Rosentau *et al.*) and calculated sea level at the location of each indicator (the spread in the response is due to spatial variations in calculated sea level). This plot shows the difference between the standard version of PaleoMIST 1.0 with 2500 year time steps, and a model run where the load is linearly interpolated to 500 year time steps. The higher temporal resolution better captures the deglaciation, and the calculated sea level is closer to the observations.

1.4 MIS 3 ice scenarios

We have provided a comparison of the scenarios both with and without Hudson Bay being ice covered in the second report. The available sea level data for MIS 3 cannot distinguish which scenario is more likely. The calculated deglacial sea level in Canada has some sensitivity to the scenarios, providing slightly better fits with the ice free scenario. Given all other uncertainties associated with the MIS 2 glaciation, it is unlikely that the deglacial sea level can provide meaningful insight on which scenario is true.

2 Reviewer #1

The manuscript “A new global ice sheet reconstruction for the past 80.000 years” presents the first ice sheet reconstruction for the last 80 ka that is independent of far-field records. A simple ice sheet model constrained by geological observations are used to reconstruct ice sheet history and volume only by near field sea level indicators. The new reconstructions (PaleoMIST 1.0) enable the authors to address a very controversial topic in paleoclimate science; the “missing ice” during LGM where there is a large discrepancy (8-28 m) between estimates of ice volume based on ice sheet reconstructions and sea level records from far-field records. In addition, they estimate a minimum and maximum MIS 3 ice extent, and calculate ice volume and find that both scenarios are significantly smaller than far-field data is suggesting. They claim that there LGM reconstruction is consistent with far-field data and furthermore highlight that $\delta^{18}O$ is not fully understood and one should be caution to use the record as ice volume indicator. In principal, this seems to be a novel approach and scientific results that potentially could be suitable for publication in Nature Communications.

We thank the reviewer for taking time to review our study.

However, I have some concerns about the study. The overall premise is that there is a mismatch between near-field geological information about ice sheet size and far-field sea level indicators. The authors furthermore write that most studies use far-field sea level data to tune ice sheet models which results in very large ice sheets to account for the sea level lowering. In the new study the authors only use near-field data and a simple ice sheet model to estimate ice volume and sea level lowering during LGM without tuning the model using the far-field data. Their max. model suggests 20% less ice volume than other tunes models, but at the same time they claim that is consistent with far-field data. How can that add-up? The starting premise was that far-field data might be erroneous and should not be used to tune models – how can it suddenly be consistent? Apparently, their data is consistent with far-field data from Barbados, Sunda Shelf and Great Barrier Reef. But on the other hand, it is inconsistent with Lambeck et al (2014) which is based on the same far-field data but corrected for isostatic effects and it indicate +130 m lower sea level at LGM. Is the GIA correction the problem? If so, it should be elaborated.

Response: The -130 m value proposed by Lambeck et al (2014) is likely the result of the way that they constructed the ice sheets. They started from an assumed ice sheet configuration, and tuned the ice volume to improve the fit to far-field sea level. They started with an LGM value of 28 m ESL in Antarctica, which is much larger than the 10-11 m value in our model, and ended with 23-30 m after iterations (depending on the Earth model used). The gravitational effects of the larger ice sheet in Antarctica will counteract the gravitational effects of the Northern Hemisphere ice sheets, therefore also requiring more ice in the Northern Hemisphere. We have added the following text to the end of the paragraph (originally lines 129–137):

The cause the discrepancy between our results and those of Lambeck et al.[8] probably lies in the difference in the starting ice volume that they used in Antarctica. Lambeck et al.[8] started their analysis with an assumed ice volume of 28 m ESL, which is much greater than the 10 m SLE value in our reconstruction. Due to gravitational effects, more ice will need to be added to Northern Hemisphere to counteract the extra ice in Antarctica to achieve a good fit to far-field sea level, leading to an overall larger ice sheet volume.

In summary, although the study is intriguing and potentially suitable for Nature Comm., the presentation of the results and discussion is not clear and there are inconsistencies that raises concern about the overall outcome to the study. If this can be accommodated the manuscript has the potential to make a big impact

Response: We hope that the added text and additional reports outlined in section 1 of this response provides sufficient clarifications on both the development and robustness of our model.

3 Reviewer #2

The authors have presented a very novel approach to create an entirely new global ice sheet reconstruction, without relying on previously published global models. This is not an easy task. They begin to address many different issues in the manuscript - MIS3 North American ice sheet reconstruction, LGM sea level budget, comparison to delta180 and far-field sea level data. However, for publication in Nature Communications would require more work to fully support the methods and many statements/conclusions given in the paper. Without these I would not support publication. However, if the paper is revised it would make a value publication for within the journal. I have separated my comments into a series of sub sections with a theme and highlighted in italics the relevant parts of the manuscripts - with page number (line number in brackets).

Response: We thank the reviewer for taking the time to review our study. We hope that the actions highlighted in section 1 to show the comparison of the modelled sea level to the data, plus the additional high temporal resolution is sufficient to demonstrate our claims.

Abstract (and title): 'a global ice sheet reconstruction for past 80ka'; Page 2(21) 'we present a global ice sheet reconstruction that is independent of far-field records for the past 80,000 ka'. The manuscript does not present a model from 80ka to present day. The entire interval is given on Fig.2; but comparison to far-field sea level is up to 15 ka (Fig.3) and it is stated Page 3(57) 'this time resolution is insufficient to model near-field Holocene sea-level changes'. How do you justify the presented model from 15 ka to present? If the model is not viable to model near-field sea level (due to time resolution); have you produced a simulation that is not suitable to be used for 30 ka to present day near-field sea level modelling?

Response: In the reports highlighted in section 1, we now show how our modeled sea level compares to paleo-sea level data in a variety of locations, both in glaciated regions of North America and Europe, some intermediate areas in North America and northern Russia, and far field regions in a variety of locations, including southeastern Asia. We have also included the results of a 500 year resolution simulation (where the ice load is linearly interpolated between the original 2500 year time steps), which supports our assertion that the deglacial sea level sea level is overestimated. The 500 year resolution simulation provides a fit that is much closer to observations. As mentioned above, the 2500 year time step was chosen due to technical and practical reasons. In addition, we were waiting for new margin reconstructions (e.g. [9, 10]) to become available to refine the reconstruction. A higher resolution version with a more realistic deglaciation is planned for the future.

Sea level model: You do not use a version of the code which includes with rotational feedback to treatment of retreating marine-based ice (Page 13 (169) . These are both important processes that should be included for sea-level modelling. Can you comment on this?

Response: The version of SELEN that we used for this study does include rotational feedback, and it will adjust the load if the changes in sea level causes it to be above the flotation point. We have added these points to the sentence originally on page 14:

The GIA was calculated using SELEN, which solves the sea level equation, including accounting for shoreline migration, adjustments for grounding line position, and rotational feedback[11, 12, 13].

Page 143 (178): ‘The GIA was calculated to a spherical harmonic degree of 256...’ ‘The sea-level change and earth deformation was calculated on a 1 degree grid’. If the GIA model is run at 256 (~70 km); how do you convert to a 1 degree (~110km)? This would involve some software for remodelling? projection? Is 1deg this the resolution of ICESHEET used when producing the input ice sheet reconstruction?

Response: The calculated sea level was determined directly from the output Stokes coefficients. This was then fed back into the ICESHEET program, which uses a 5 km projected grid, via Delaunay triangulation. We have added this to the first paragraph of the methods:

The sea-level change and Earth deformation was calculated on a 1 degree grid directly from the output Stokes coefficients to be input back into the ice sheet reconstruction. Since the ice sheet calculation was done on a 5 km grid, the 1 degree grid was projected and interpolated using Delaunay triangulation[14, 15].

The following sentence was added to the second paragraph of the methods:

The ice sheets were calculated on a projected coordinate system at 5 km resolution. For input into SELEN, the output ice thickness was converted into a hexagonal grid[16] with a resolution of roughly 66 km, which is sufficient for calculating to spherical harmonic degree 256.

Earth model: Page13(173):‘This set of Earth model parameters is within the range of values used in Eurasian GIA studies’. Page 6(121) ‘the GIA response in areas covered by the Eurasian ice sheet are not sensitive to changes in the lower mantle viscosity’. Why do you adopt the reference of values from the Eurasian ice sheet for the lower mantle, if you have determined (no evidence presented) it is not sensitivity to constrain this parameter?

Response: After creating the report that compares the lower mantle viscosity values from 10^{21} - 10^{23} Pa s, it is clear that this statement is incorrect (see also discussion in section 1, above). The calculated sea level in Europe is sensitive to this parameter. We have now included a figure in the supplementary material to show this. Much like the region covered by the Laurentide Ice Sheet, using a higher lower mantle viscosity ($\geq 4 \times 10^{22}$ Pa s) produces a better fit. However, the Eurasian ice sheet should still not significantly perturb the lower mantle[17], so this is likely related to the response from the North American ice sheets[7]. We have removed the sentence (originally at line 121–122) and replaced it with:

The GIA response for areas covered by the Eurasian ice sheet also fit better with our chosen mantle viscosity (4×10^{22} Pa s) than with lower values. Since the GIA response of the Eurasian ice sheet is relatively insensitive to the lower mantle[17], this response is likely influenced by the GIA signal from the North American ice complex[7].

The main focus of the paper is on the sensitivity changes in the North American ice sheet, does it not follow to use an earth model preferred by the North American ice sheet (or GIA-based studies of this ice sheet). Steffan and Wu, 2011 the average value for lower mantle (Table3 in paper) is closer to $\sim 2 \times 10^{22}$ Pa s.

Response: Actually, at the start of the tuning process, we used a variety of lower mantle viscosity values (initially defaulting at a value of 4×10^{21} Pa s). After a few iterations, it became clear to maximize the ice volume in Hudson Bay, using a higher lower mantle viscosity was necessary. Looking through the text about the tuning of the reconstruction (last paragraph of the methods), this step was not mentioned in the main text, but only in figure 2 of the supplementary material. We have added these details to this paragraph (inserted after the discussion on varying the shear stress, on line 322 in the original submission):

After some initial tuning, we tested the impact of varying the lower mantle viscosity on postglacial sea level in Hudson Bay. The initial Earth model that was utilized had a viscosity of 4×10^{21} Pa s, but the modelled sea level was much higher than the observations. Tests using different lower mantle viscosity values revealed that values approaching 10^{23} Pa s reduced the postglacial sea level. Using a higher viscosity value made it possible to have a larger ice load in the center of the Laurentide Ice Sheet, and therefore maximize ice volume, so a value of 4×10^{22} Pa s was chosen for subsequent tuning.

Page6 (118) ‘we perform tests using a variety of lower mantle viscosity values’. What range of models? what tests? - Did you run all simulations for a suite of models? you state you use models that are from a far-field study. However, as above, your model development is based on near-field evidence only, so why adopt an earth model from far-field reconstruction?

Response: As mentioned in the text (line 117-119) we varied the lower mantle viscosity in order to maximize ice volume in the Hudson Bay region (as in the previous response). We tested a range of lower mantle viscosity values, ranging between 10^{21} and 10^{23} Pa s, which was the range found to produce good results in far field regions[8]. We have now added this range to the the sentence:

We performed tests using a variety of lower mantle viscosity values, specifically the best fitting values for far-field indicators[8] (10^{21} – 10^{23} Pa s).

Time step: Page 3(56) ‘Since our ice sheet reconstruction focuses on general ice sheet configuration changes over the past 80 ka, we use a relevantly coarse time step of 2.5 ka...it is insufficient to model..Holocene... rapid sea-level rates’. There is a rapid change in the RSL rate ~20 ka with the inflection at the onset of the deglaciation. Is 2.5 ka sufficient to capture this?

Response: We are not exactly sure what this comment is directly referring to. If it is referring to globally rapid sea level rise events (like Meltwater Pulse 1a), then obviously the 2.5 ka resolution, which is a smoothed representation of the glacial history, will not capture this. This sentence explicitly refers to near field Holocene records, which show rapid sea level fall after deglaciation. As demonstrated in the high temporal resolution simulation detailed in section 1, using the higher temporal resolution does reduce the misfit. The far field records are most sensitive to ice volume changes, and are not as sensitive to a 2500 year versus a linearly interpolated 500 year time step. Please refer to the report showing the comparison of the high and low resolution simulations.

Page2(23) ‘thickness is controlled by near-field sea level indicators’. Page 14 (194) ‘Due to the 2.5ka time step in this reconstruction, the post-glacial sea level in the core areas of the ice sheet is overestimated’. Page 19 (296) ‘the basal shear stress was adjusted ... to match the ice flow direction’. Initially it is stated that near-field sea level is used to control thickness, but the time step is too coarse to model in the core areas. However, in the full method description you state the flow lines are used? Please be clear. I understand and it is approach to use the coarse resolution when there is little data, but how have you evaluated the impact on the modelling? why not reduce later in the reconstruction?

Response: In order to clarify our choice for this 2500 year time step, we have added the following paragraph to the methods section after the first paragraph:

A technical limitation of the version of SELEN we use is that it requires a constant time step, so during the development of the reconstruction we chose not to pursue a more refined time steps during the deglaciation when there is more information, as it would substantial increase the computing time for our exploratory study. This is why this reconstruction has a relatively coarse 2500 year time step. The goal of our reconstruction was to develop the framework to generate global ice sheet reconstructions and then to evaluate general ice sheet change for the past 80,000 years. To test the robustness of the ice sheet reconstruction, we did run a test where we linearly interpolated the ice load to 500 year time steps (see supplementary figure). In near field areas, the 500 year time step simulation has a much better fit to the data. In far field regions where sea level is dominantly affected by ice volume changes, the response is essentially the same. A version of SELEN with variable time steps is in development, and a more refined ice sheet reconstruction is being planned to utilize recently published ice sheet margin reconstructions[9, 10].

Response: Both the sea level data and ice flow direction data were used to adjust the ice sheet reconstructions, specifically the margins, but also the shear stress, especially during the period before the LGM (e.g. see lines 214–217 and 274–275 in the original submission). However, we used the flow direction only to ensure the surface gradient (and therefore flow direction) is correct.

LGM sea level budget. The authors have presented one reconstruction which captures the upper error limits of the far-field sea level data (Fig.3). This is great and not an easy task. However, to fully evaluate this, also requires evaluation of the period post LGM. From Fig3, the authors only evaluate the model to the far-field sea level up to ~15 ka; why disregard the entire record?. The model underpredicts the delta 180 over the deglaciation (Fig.2) and GBR data. Page 7 (136): ‘Given our reconstruction does not include components.... the ice volumes are likely overestimated’ From this statement, does it imply that your results would be ~3-4m lower, increasing this underprediction?

Response: We have addressed this in the steps described in section 1 of this response, specifically to show the model-data comparison for all of the locations that we used. This includes far field records in places like Barbados, Tahiti and Huon Peninsula where continuous records of far field sea level through the deglaciation exist. Since our focus was primarily to show the lowstand at the LGM, in the paper we only showed data from locations where there are constraints for the LGM lowstand. Since the $\delta^{18}\text{O}$ record does not directly record sea level changes, we have no expectation than our curves fit it except in a qualitative sense (especially since the $\delta^{18}\text{O}$ data are not consistent during MIS 3). There is a discrepancy between our modelled sea level and the sea level indicators at the Great Barrier Reef in Mackay, but not at Cairns. The difference between these two sites cannot be explained by the GIA response of our model, since the calculate sea level is almost identical at the two sites. There was no discussion of this offset in the original study[18], so we cannot elaborate. Note that we did not use these data to tune the ice sheet reconstructions, and given the scatter in the data, the vertical uncertainties may be underestimated. For the last point, since other sources of water volume, such as mountain glaciers, ice caps and thermal contraction are not included, this implies that if our ice sheet reconstruction exactly fits the sea level data, the sum of the other sources would increase the sea level drop, thus making an overprediction as we state (although from Simms et al[19], this is likely not more than 3–4 m of sea level equivalent).

Near-field landform data/Ice sheet reconstruction: The approach of developing a complete new ice sheet reconstruction from scratch (for want of a better phrase) is a complicated task and the work the authors present is good. The development of a new approach to do this, given the issues (circularity) with using far-field data is great. However, there are a number of very general statements (both in the main paper and Method) of how ice sheet reconstruction is evaluated or developed in comparison to near field data for which evidence is not provided. The method describes the use of ice flow direction: How is this elated? do you model ice streams? a GIA model does not have ice flow (or the plastic ice sheet model).

Response: Thank you, the framework for this study has been developed over the course of the past 10

years, so it indeed has been a long haul. With the regards to the near field data sea level data, we have described this in section 1. In regards to the flow directions, it is assumed that the ice flow direction will follow the topographic gradient of the surface of the ice sheet. To address this, we have added a paragraph to the methods section.

For the flow direction indicators, we assume that the ice sheet flow will be in the direction of greatest ice sheet surface gradient[20, 21]. Thus, by examining geological indicators of ice flow direction, it is possible to evaluate if the ice sheet reconstruction is realistic. Specifically for the pre-LGM Laurentide Ice Sheet, we made adjustments to the ice margin and basal shear stress to ensure that the ice sheet surface gradient followed know flow direction shifts[22]. While this does not directly constrain ice thickness or volume (and is therefore non-unique), it ensures that for our chosen combination of ice margin and shear stress, that geometry remains internally consistent with geological constraints.

The Eurasian ice sheet reconstruction which incorporate the DATED-1 information, is great. The constraints in paper DATED-1 as the authors allude to, do not extent into MIS4. However, Lambeck2010 ‘The Scandinavian ice sheet: MIS4 to the end of the LGM’ reproduce an reconstruction, using evidence from earlier MIS4. How does your reconstruction compare during this earlier period.

Response: Since Lambeck et al.’s model is not publicly available, we can only make a qualitative comparison with the figures in that paper. We add a sentence (after line 225 in the original manuscript):

Compared to a previous reconstruction[23], the ice extent is less at 40 000–50 000 yr BP, but similar to the minimal extent at 35 000 yr BP.

Near-field sea level data: In the SOM the modelled RSL is compared to a very limited number of Holocene sea level data? There is no reference to these results in the paper. What near-field sea level data was used in the development of the North American ice sheet. If it is only the MIS4/3 data, how is the 30 ka to present day evaluated (without using the far-field data). I would consider it necessary to evaluate (even using some simple statistics) the fit of the rsl predictions to your LGM far-field sea level data.

Response: A thorough comparison of modelled sea level to the data are presented in detail in section 1 of this review.

4 Reviewer #3 (Martin Margold)

The manuscript ‘A new global ice sheet reconstruction for the past 80 000 years’ by Gowan et al. is a much needed study that attempts to shed light on the evolution of the ice sheets of the last glaciation from a global perspective. The study is based on glacial-isostatic adjustment (GIA) methods and I assume the other approached reviewers have full insight into that, which I don’t. With my main expertise in palaeoglaciological reconstructions based on empirical data, I will focus my review on the empirical information on ice sheet margins that the authors use as one of the main inputs into their reconstruction.

Response: We thank Dr. Margold for reviewing our paper.

The authors utilise previously published information for the Eurasian ice sheets and carry out their own literature survey to source information for the remaining ice masses (described at lines 221-294). I generally find their described approach sound and well-informed. One point that I would raise is that given the information availability, some better documented minor

ice masses, such as the Patagonian Ice Sheet or the Brooks Range, got included, while the glaciation of Siberia, that at MIS 4 likely contained a sea level equivalent of more than 1 m, was not included. But I understand that this is likely because of the lack of information for that region.

Response: The Brooks Range was included because it was part of the base margin reconstruction that we used by Dyke[24]. At an early stage, we considered including Siberian and other mountain based ice caps, but after finding the minor amount of ice volume that was in the much larger Patagonia ice sheet (<1 m), we did not think it was worth considering at this time. If you are referring to the eastern Siberian Shelf ice sheet, we are unaware of any studies that confidently place its existence with the Late Pleistocene.

For the Laurentide Ice Sheet, the history of which is also poorly understood before the LGM, the approach of using the Heinrich events as a proxy for the position and timing of the eastern margin is probably sound, even though it hinges the history of the ice sheet on a set of events that we still understand rather poorly.

Response: We agree, we hope that this study will be stepping stone to encourage others to try to find evidence that will better constrain the ice sheet history prior to the LGM.

The authors refer briefly to the recent publication of Batchelor et al. (2019) that aimed to compile the available information on the extents of Northern Hemispheric ice sheets at selected time slices / periods prior to the LGM (the paper contains 6 time slices relevant for this study). It would be interesting to see even a brief discussion of matches and mismatches between the outlines of Batchelor et al. and those of this reconstruction.

Response: We have included the following figures comparing the MIS 3 and MIS 4 extents from Batchelor et al.[25] in the supplementary material.

Figure 7: Comparison of the MIS 3 minimum extent margin reconstruction by Batchelor *et al.*[25], and PaleoMIST 1.0 for the North America and Eurasian ice sheets. The minimum extent in the Batchelor *et al.* reconstruction is set to 45000 yr BP in their “best estimate” margin set. The minimal ice scenario (with an ice free Hudson Bay) is set to be 42500 yr BP in PaleoMIST 1.0, while the minimum extent of Eurasia is set to 40000 yr BP.

Figure 8: Comparison of the MIS 4 maximum extent margin reconstruction by Batchelor *et al.*[25], and PaleoMIST 1.0 for the North America and Eurasian ice sheets. The margin displayed here for *et al.* is their “best estimate” margin set. The PaleoMIST 1.0 margin corresponds to 60000 yr BP

I hope that the authors plan to make the modelled results available in a way that other GIA reconstructions, such as that of Lambeck et al. (2017) or Peltier et al. (2015), have been. A short section on data availability could be added for that.

Response: The data have been uploaded to Pangaea, and the code is available on Github. The data availability section was on lines 600–604 of the original document.

To conclude, I believe that this will be a widely used resource for many in the Quaternary / palaeoclimate community. The manuscript is well structured and written in a good style. I recommend it for publication in Nature Communications.

Response: Thank you. I hope that this is the start of a new generation of ice sheet reconstructions that provide the flexibility to be useful for the whole community.

L 241 the error ranges become

Response: Fixed.

L 243 the chronology was set

Response: Fixed

L 282-283 ‘The MIS 4 glacial maximum for the Cordilleran Ice Sheet happened at about 55 000 yr BP, which is somewhat later than what we set the Laurentide Ice Sheet’. This statement needs to be either referenced or briefly explained

Response: We have modified the sentence to be as follows:

The MIS 4 glacial maximum for the Cordilleran Ice Sheet happened at about 55 000 yr BP[26, 27], which is somewhat later than what we set the Laurentide Ice Sheet (which was based on the timing of Heinrich Event 6[28]).

5 Other changes

The abstract has been rewritten to adhere to the standards of Nature Communications (*i.e.* 150 words and without references).

We present a global ice sheet reconstruction for the past 80 000 years, called PaleoMIST 1.0, constructed independently of far-field sea level and $\delta^{18}\text{O}$ proxy records often used to constrain ice volume. This allows us to assess the “missing ice problem”, the apparent 8-28 m lower far-field sea level during the Last Glacial Maximum (LGM, 26 000-19 000 years before present) than from ice sheet reconstructions, and the reliability of proxy records as indicators of sea level. Our reconstruction is compatible with LGM far-field sea-level records without requiring extra ice volume, thus solving the missing ice problem. Neither of two presented scenarios for the pre-LGM period match proxy based sea level reconstructions for Marine Isotope Stage 3, indicating the relationship between oceanic $\delta^{18}\text{O}$ and sea level may be more complex than assumed.

Subsection headers have also been added to the methods section.

Best Regards,
Evan J. Gowan (on behalf of the authors)

References

- [1] Gowan, E. J. *et al.* ICESHEET 1.0: A program to produce paleo-ice sheet models with minimal assumptions. *Geoscience Model Development Discussions* -, – (2016). doi:10.5194/gmd-2016-9. In review.
- [2] Gowan, E. J. Comparison of the PaleoMIST 1.0 ice sheet margins, ice sheet and paleo-topography reconstruction with paleo sea level indicators (2020). doi:10.5281/zenodo.4061594. URL <https://doi.org/10.5281/zenodo.4061594>.
- [3] James, T. S., Gowan, E. J., Wada, I. & Wang, K. Viscosity of the asthenosphere from glacial isostatic adjustment and subduction dynamics at the northern Cascadia subduction zone, British Columbia, Canada. *Journal of Geophysical Research: Solid Earth* **114**, B04405 (2009). doi:10.1029/2008JB006077.
- [4] Peltier, W. R., Argus, D. F. & Drummond, R. Space geodesy constrains ice age terminal deglaciation: The global ICE-6G.C (VM5a) model. *Journal of Geophysical Research: Solid Earth* **120**, 450–487 (2015). doi:10.1002/2014JB011176.
- [5] Lambeck, K., Purcell, A. & Zhao, S. The North American Late Wisconsin ice sheet and mantle viscosity from glacial rebound analyses. *Quaternary Science Reviews* **158**, 172–210 (2017). doi:10.1016/j.quascirev.2016.11.033.
- [6] Vacchi, M. *et al.* Postglacial relative sea-level histories along the eastern Canadian coastline. *Quaternary Science Reviews* **201**, 124–146 (2018). doi:10.1016/j.quascirev.2018.09.043.
- [7] Wu, P. Sensitivity of relative sea levels and crustal velocities in Laurentide to radial and lateral viscosity variations in the mantle. *Geophysical Journal International* **165**, 401–413 (2006). doi:10.1111/j.1365-246X.2006.02960.x.
- [8] Lambeck, K., Rouby, H., Purcell, A., Sun, Y. & Sambridge, M. Sea level and global ice volumes from the Last Glacial Maximum to the Holocene. *Proceedings of the National Academy of Sciences* **111**, 15296–15303 (2014). doi:10.1073/pnas.1411762111.

- [9] Dalton, A. S. *et al.* An updated radiocarbon-based ice margin chronology for the last deglaciation of the North American Ice Sheet Complex. *Quaternary Science Reviews* **234**, 106223 (2020). doi:<https://doi.org/10.1016/j.quascirev.2020.106223>.
- [10] Davies, B. J. *et al.* The evolution of the Patagonian Ice Sheet from 35 ka to the present day (PATICE). *Earth-Science Reviews* **204**, 103152 (2020). doi:<https://doi.org/10.1016/j.earscirev.2020.103152>.
- [11] Spada, G. & Stocchi, P. SELEN: A Fortran 90 program for solving the “sea-level equation”. *Computers & Geosciences* **33**, 538–562 (2007). doi:[10.1016/j.cageo.2006.08.006](https://doi.org/10.1016/j.cageo.2006.08.006).
- [12] de Boer, B., Stocchi, P. & Van De Wal, R. A fully coupled 3-D ice-sheet-sea-level model: algorithm and applications. *Geoscientific Model Development* **7**, 2141–2156 (2014). doi:[10.5194/gmd-7-2141-2014](https://doi.org/10.5194/gmd-7-2141-2014).
- [13] de Boer, B., Stocchi, P., Whitehouse, P. L. & van de Wal, R. S. Current state and future perspectives on coupled ice-sheet – sea-level modelling. *Quaternary Science Reviews* **169**, 13–28 (2017). doi:<https://doi.org/10.1016/j.quascirev.2017.05.013>.
- [14] Shewchuk, J. R. Triangle: Engineering a 2D quality mesh generator and Delaunay triangulator. In Lin, M. C. & Manocha, D. (eds.) *Applied Computational Geometry Towards Geometric Engineering*, 203–222 (Springer Berlin Heidelberg, Berlin, Heidelberg, 1996). doi:<https://doi.org/10.1007/BFb0014497>.
- [15] Wessel, P., Smith, W. H., Scharroo, R., Luis, J. & Wobbe, F. Generic mapping tools: improved version released. *Eos, Transactions American Geophysical Union* **94**, 409–410 (2013). doi:[10.1002/2013EO450001](https://doi.org/10.1002/2013EO450001).
- [16] Tegmark, M. An icosahedron-based method for pixelizing the celestial sphere. *The Astrophysical Journal Letters* **470**, L81–L84 (1996). doi:<https://doi.org/10.1086/310310>.
- [17] Steffen, H. & Kaufmann, G. Glacial isostatic adjustment of Scandinavia and northwestern Europe and the radial viscosity structure of the Earth’s mantle. *Geophysical Journal International* **163**, 801–812 (2005). doi:[10.1111/j.1365-246X.2005.02740.x](https://doi.org/10.1111/j.1365-246X.2005.02740.x).
- [18] Yokoyama, Y. *et al.* Rapid glaciation and a two-step sea level plunge into the Last Glacial Maximum. *Nature* **559**, 603 (2018).
- [19] Simms, A. R., Lisiecki, L., Gebbie, G., Whitehouse, P. L. & Clark, J. F. Balancing the Last Glacial Maximum (LGM) sea-level budget. *Quaternary Science Reviews* **205**, 143–153 (2019). doi:[10.1016/j.quascirev.2018.12.018](https://doi.org/10.1016/j.quascirev.2018.12.018).
- [20] Reeh, N. A plasticity theory approach to the steady-state shape of a three-dimensional ice sheet. *Journal of Glaciology* **28**, 431–455 (1982).
- [21] Fisher, D., Reeh, N. & Langley, K. Objective reconstructions of the Late Wisconsinan Laurentide Ice Sheet and the significance of deformable beds. *Géographie Physique et Quaternaire* **39**, 229–238 (1985).
- [22] Gauthier, M. S. *et al.* The subglacial mosaic of the Laurentide Ice Sheet; a study of the interior region of southwestern Hudson Bay. *Quaternary Science Reviews* **214**, 1–27 (2019). doi:[10.1016/j.quascirev.2019.04.015](https://doi.org/10.1016/j.quascirev.2019.04.015).
- [23] Lambeck, K., Purcell, A., Zhao, J. & Svensson, N.-O. The Scandinavian Ice Sheet: from MIS 4 to the end of the Last Glacial Maximum. *Boreas* **39**, 410–435 (2010). doi:[10.1111/j.1502-3885.2010.00140.x](https://doi.org/10.1111/j.1502-3885.2010.00140.x).
- [24] Dyke, A. S. An outline of North American deglaciation with emphasis on central and northern Canada. In Ehlers, J., Gibbard, P. L. & Hughes, P. D. (eds.) *Quaternary Glaciations—Extent and Chronology - Part II: North America*, Developments in Quaternary Science, 373–424 (Elsevier, 2004). doi:[10.1016/S1571-0866\(04\)80209-4](https://doi.org/10.1016/S1571-0866(04)80209-4).

- [25] Batchelor, C. L. *et al.* The configuration of Northern Hemisphere ice sheets through the Quaternary. *Nature communications* **10**, 1–10 (2019). doi:10.1038/s41467-019-11601-2.
- [26] Ward, B. C., Bond, J. D. & Gosse, J. C. Evidence for a 55–50 ka (early Wisconsin) glaciation of the Cordilleran ice sheet, Yukon Territory, Canada. *Quaternary Research* **68**, 141–150 (2007). doi:10.1016/j.yqres.2007.04.002.
- [27] Mathewes, R. W., Lian, O. B., Clague, J. J. & Huntley, M. J. W. Early Wisconsinan (MIS 4) glaciation on Haida Gwaii, British Columbia, and implications for biological refugia. *Canadian Journal of Earth Sciences* **52**, 939–951 (2015). doi:10.1139/cjes-2015-0041.
- [28] Andrews, J. T. & Voelker, A. H. “Heinrich Events” (& sediments): A history of terminology and recommendations for future usage. *Quaternary Science Reviews* **187**, 31–40 (2018). doi:10.1016/j.quascirev.2018.03.017.

Reviewer comments, second round –

Reviewer #1 (Remarks to the Author):

I thank the authors for considering my previous comments about the manuscript (Reviewer #1). In the revised version they have clarified all my concerns and I can recommend publication with only minor revision.

Minor things:

Line 29: GIA should be defined the first time used.

Line 56-57: In Greenland there is also evidence for a moderate ice extent during MIS 3 (e.g. Søndergaard et al 2019, JQS; and references therein). The data from North Greenland would support the minimum MIS 3 scenario.

Line 61: change "glacial isostatic adjustment" modelling to "GIA" modelling.

Line 99: SLE has not been defined.

Line 117: Eurasian ice sheet or sheets?

Line 120: North American ice complex – Maybe change to Laurentide and Cordilleran ice sheets? or is Greenland ice sheet included in the North American ice complex?

Line 140: Reword – first part of sentence is confusing. During MIS-3, since...

Line 252: Eurasian ice sheet or sheets?

Reviewer #2 (Remarks to the Author):

The authors have addressed all my comments raised in the previous review. One point that was raised relating to a comparison of the reconstruction (PaleoMIST 1.0) to a range of sea level indicators. The authors have completed an outstanding amount of additional evaluation of the reconstruction; which has been summarised within the main document and provided as an external link. Given the huge amount of work that will have gone into this, I think it would be also interesting to see this published in greater detail in a follow up publication.

The paper is an interesting and topical publication on the topic of MIS4 -MIS3 ice sheet extent and the construction of global ice sheet reconstructions; without the additional 2nd point made in the paper addressing the missing ice at the LGM. I have one follow up question regarding the definition of 'missing ice' and reference to the Lambeck study(ref 29); which I think is a clarification that could be added to the paper with some explanation.

It is correct as the author state Lambeck ESL reconstruction is ~ 130m due to a larger AIS.

The missing ice problem (as defined in Simms et al., 2019) is resolving the mismatch between the sum of the total ESL contribution from the global ice sheets: 116.4/113.9m (Simms et al.2019 when estimated from a range of published model) and the estimated LGM lowstand of -130m (Fig. 4 Lambeck). The 130m estimate from Lambeck study (and other global ice sheet reconstructions (i.e ICE*G family) is dictate by the estimates of the observed sea level from data records, far-field records. When these records are not adopted (or used, as the authors methods) then their reconstruction is of a similar magnitude to the range of results published in Simms et al., 2019. If an LGM lowstand at a far-field site was correct/accurate (i.e the data as published in Lambeck, Fig4); how would the authors reconstruction address this? It seems that the missing ice issue has been resolved by a 'reinterpretation' of sea level indicators and the size of the error's bars.

Reviewer #3 (Remarks to the Author):

I have now been through the revised version of the manuscript and I do not have any further comments to it. I would like to commend the authors for what I see as an important contribution.

Minor issues with the text:

P4 L79 The timing is targeted or timings are targeted

P5 L87 Check the wording, there's a definite article forgotten in the text

P13 L177 are the same as used in the...

P16 L246 known

P16 L253 The main characteristics of ... were OR the main characteristic of ... was

P17 L268 The East Antarctic Ice Sheet

Best regards,
Martin Margold

A new global ice sheet reconstruction for the past 80 000 years – response to second round of comments

Evan J. Gowan^{1,2}, Xu Zhang^{1,3}, Sara Khosravi⁴, Alessio Rovere², Paolo Stocchi⁵,
Anna L. C. Hughes^{6,7}, Richard Gyllencreutz⁸, Jan Mangerud⁷, John-Inge Svendsen⁷
& Gerrit Lohmann^{1,2}

November 16, 2020

1. Alfred Wegener Institute, Helmholtz Center for Polar and Marine Research, Bremerhaven, Germany
2. MARUM, University of Bremen, Bremen, Germany
3. Key Laboratory of Western China's Environmental Systems (Ministry of Education), College of Earth and Environmental Science, Centre for Pan Third Pole Environment (Pan-TPE), Langzhou University, Langzhou, China
4. Alfred Wegener Institute, Helmholtz Center for Polar and Marine Research, Potsdam, Germany
5. NIOZ, Texel, Netherlands
6. Department of Geography, University of Manchester, Manchester, United Kingdom
7. Department of Earth Science, University of Bergen and Bjerknes Centre for Climate Research, Bergen, Norway
8. Department of Geological Sciences, Stockholm University, Stockholm, Sweden

1 Overview

First off, we would like to thank the three reviewers taking the time to review our paper. We recognize the difficulties during these trying times. We appreciate their recognition of the amount of work that went into the revised paper, including the sea level analysis. Here, we summarize the main steps we have taken to address the comments. The original comments from the reviewers are in italics, our response is in standard font, and changes to the text in the manuscript are shown in blue.

2 Reviewer #1

I thank the authors for considering my previous comments about the manuscript (Reviewer #1). In the revised version they have clarified all my concerns and I can recommend publication with only minor revision.

Response: We thank the reviewer for taking the time to review our paper.

Line 29: GIA should be defined the first time used.

Response: We have now defined “glacial isostatic adjustment”

Line 56-57: In Greenland there is also evidence for a moderate ice extent during MIS 3 (e.g. Søndergaard et al 2019, JQS; and references therein). The data from North Greenland would support the minimum MIS 3 scenario.

Response: Thank you for the reference. This paper came out after we had finalized our ice sheet margin reconstructions, so it did not impact our model. In a future iteration, we will consider it, which will cause a minor reduction in the ice extent in northwestern Greenland compared to our current model. This particular section (line 55-57) is specifically referring to the Hudson Bay area. We have added a reference to Søndergaard et al 2019 in the methods section where we discuss Greenland (lines 316–320 in the original manuscript).

The northwestern Greenland Ice sheet also may be seen a similar retreat during MIS 3[1].

Line 61: change “glacial isostatic adjustment” modelling to “GIA” modelling.

Response: The acronym has replaced the text.

Line 99: SLE has not been defined.

Response: We have defined SLE:

SLE - the equivalent amount of water, in metres, that would cover the ocean surface if the ice melted

Line 117: Eurasian ice sheet or sheets?

Response: It has been changed to “ice sheets”.

Line 120: North American ice complex – Maybe change to Laurentide and Cordilleran ice sheets? or is Greenland ice sheet included in the North American ice complex?

Response: This has been changed to “North American ice sheets”, which does include Greenland.

Line 140: Reword – first part of sentence is confusing. During MIS-3, since...

Response: We have reworded this sentence to be:

Since the Eurasian ice sheets were generally restricted to mountainous areas during the middle of MIS 3[2, 3], fluctuations in global sea level were controlled almost exclusively by the Laurentide Ice Sheet (Fig. 2).

Line 252: Eurasian ice sheet or sheets?

Response: This has been changed to “ice sheets” here and everywhere else in the paper.

3 Reviewer #2

The authors have addressed all my comments raised in the previous review. One point that was raised relating to a comparison of the reconstruction (PaleoMIST 1.0) to a range of sea level indicators. The authors have completed an outstanding amount of additional evaluation of the reconstruction; which has been summarised within the main document and provided as an external link. Given the huge amount of work that will have gone into this, I think it would be also interesting to see this published in greater detail in a follow up publication.

Response: We thank the reviewer for looking at our paper and acknowledging our efforts to create the sea level indicator documents. We intend to create a followup paper with a higher temporal resolution ice sheet reconstruction for the deglacial period, as well as to include more sea level datasets.

The paper is an interesting and topical publication on the topic of MIS4 -MIS3 ice sheet extent and the construction of global ice sheet reconstructions; without the additional 2nd point made in the paper addressing the missing ice at the LGM. I have one follow up question regarding the definition of 'missing ice' and reference to the Lambeck study(ref 29); which I think is a clarification that could be added to the paper with some explanation.

*It is correct as the author state Lambeck ESL reconstruction is \130m due to a larger AIS. The missing ice problem (as defined in Simms et al., 2019) is resolving the mismatch between the sum of the total ESL contribution from the global ice sheets: 116.4/113.9m (Simms et al.2019 when estimated from a range of published model) and the estimated LGM lowstand of -130m (Fig. 4 Lambeck). The 130m estimate from Lambeck study (and other global ice sheet reconstructions (i.e ICE*G family) is dictate by the estimates of the observed sea level from data records, far-field records. When these records are not adopted (or used, as the authors methods) then their reconstruction is of a similar magnitude to the range of results published in Simms et al., 2019. If an LGM lowstand at a far-field site was correct/accurate (i.e the data as published in Lambeck, Fig4); how would the authors reconstruction address this? It seems that the missing ice issue has been resolved by a 'reinterpretation' of sea level indicators and the size of the error's bars.*

Response: We agree, an alternative explanation is that the far-field records are not necessarily representative of the global average sea level and that the very definition of “global” sea level is model dependent. We have now included an additional paragraph to describe this alternative explanation.

An alternative explanation is that the far-field records that show LGM sea level values below -120 m are not representative of the global average sea level. In our model at the LGM (Fig. 3), much of the Southern Hemisphere ocean, as well as coastal regions of southern and southeastern Asia have calculated sea level below the global average. The global average sea level is only represented in relatively narrow regions, such as in northern South America, western Africa and Australia (*e.g.* Bonaparte Gulf and Cairns, Fig 3). Our model also has somewhat lower (by several meters) sea level in some far field regions due to our choice of a lower mantle viscosity that is higher than other ice reconstruction studies[4]. In this case, it is not so much that our model fixes the “missing ice problem”, but that the definition of “global” sea-level is model dependent. We recommend that GIA based ice sheet reconstruction work should not target a specific ice volume or “global” sea level lowstand value, but consider each location as independent observations of relative sea level within a framework of sites around the world.

4 Reviewer #3 (Martin Margold)

I have now been through the revised version of the manuscript and I do not have any further comments to it. I would like to commend the authors for what I see as an important contribution.

Response: We thank Dr. Margold for reviewing our paper.

P4 L79 The timing is targeted or timings are targeted

Response: Changed to “is targeted”

P5 L87 Check the wording, there's a definite article forgotten in the text

Response: We removed the spurious “the” that was in the sentence.

P13 L177 are the same as used in the...

Response: We added the missing “in”.

P16 L246 known

Response: Fixed it to say “known” instead of “know”.

P16 L253 The main characteristics of... were OR the main characteristic of... was

Response: We changed to be “characteristics”.

P17 L268 The East Antarctic Ice Sheet

Response: Fixed it to say “Antarctic” rather than “Antarctica”.

Best Regards,
Evan J. Gowan (on behalf of the authors)

References

- [1] Søndergaard, A. S., Larsen, N. K., Olsen, J., Strunk, A. & Woodroffe, S. Glacial history of the greenland ice sheet and a local ice cap in qaanaaq, northwest greenland. *Journal of Quaternary Science* **34**, 536–547 (2019). doi:<https://doi.org/10.1002/jqs.3139>.
- [2] Helmens, K. F. The Last Interglacial–Glacial cycle (MIS 5–2) re-examined based on long proxy records from central and northern Europe. *Quaternary Science Reviews* **86**, 115–143 (2014). doi:[10.1016/j.quascirev.2013.12.012](https://doi.org/10.1016/j.quascirev.2013.12.012).
- [3] Hughes, A. L., Gyllencreutz, R., Lohne, Ø. S., Mangerud, J. & Svendsen, J. I. The last Eurasian ice sheets—a chronological database and time-slice reconstruction, DATED-1. *Boreas* **45**, 1–45 (2016). doi:[10.1111/bor.12142](https://doi.org/10.1111/bor.12142).
- [4] Gowan, E. J. Comparison of the PaleoMIST 1.0 ice sheet margins, ice sheet and paleo-topography reconstruction with paleo sea level indicators (2020). doi:[10.5281/zenodo.4061594](https://doi.org/10.5281/zenodo.4061594). URL <https://doi.org/10.5281/zenodo.4061594>.

Reviewer comments, third round –

Reviewer #2 (Remarks to the Author):

Thank you for the additional paragraph to clarify missing ice and LGM lowstands.

I recommend publication.

Response to Referee's comments

Reviewer #2 (Remarks to the Author):

Thank you for the additional paragraph to clarify missing ice and LGM lowstands.

I recommend publication.

Response: We thank the reviewer for reviewing our paper.